# Orb2 enables rare-codon-enriched mRNA expression during *Drosophila* neuron differentiation

Rebeccah K. Stewart[1,2], Patrick Nguyen [1], Alain Laederach [3], Pelin C. Volkan [4], Jessica K. Sawyer[1,2] & Donald T. Fox [1,2]✉

Regulation of codon optimality is an increasingly appreciated layer of cell- and tissue-specific protein expression control. Here, we use codon-modified reporters to show that differentiation of *Drosophila* neural stem cells into neurons enables protein expression from rare-codon-enriched genes. From a candidate screen, we identify the cytoplasmic polyadenylation element binding (CPEB) protein Orb2 as a positive regulator of rare-codon-dependent mRNA stability in neurons. Using RNA sequencing, we reveal that Orb2-upregulated mRNAs in the brain with abundant Orb2 binding sites have a rare-codon bias. From these Orb2-regulated mRNAs, we demonstrate that rare-codon enrichment is important for mRNA stability and social behavior function of the metabotropic glutamate receptor (mGluR). Our findings reveal a molecular mechanism by which neural stem cell differentiation shifts genetic code regulation to enable critical mRNA stability and protein expression.

During the progression from stem cell to terminally differentiated cell, numerous molecular changes occur at the level of chromatin, transcription, and translation[1–6]. An underexplored mechanism of this progression is the regulation of biased codon usage[7–9]. Generally, codons that are rarely used in a coding genome are nonoptimal for the translation of an mRNA, whereas codons that are commonly used are optimal. Therefore, mRNAs that are enriched in rarely used codons are often both less stable and less robustly translated[10,11]. This lower expression from rare-codon-enriched mRNAs can enable optimal regulation of cell signaling or more rapid protein turnover[12–15].

Cellular context impacts the optimality of specific rare codons for protein expression[12,16–26]. An open question is how codon optimality is regulated in the context of cellular differentiation. Molecular regulators that might link differentiation to changes in codon regulation are likely to converge on processes such as regulation of translation/tRNAs[27–32] or mRNA stability[19,33–38]. We previously showed that the *Drosophila* testis and brain are distinctly capable of upregulating rare-codon expression[16], suggesting that molecular regulators that are unique to these tissues may be important in codon-dependent regulation.

In the brain, neurons provide a particularly interesting context in which regulation of codon optimality is likely coupled to cell differentiation. Recent studies have found that neural tissues have important codon-dependent regulation, with increased stability of transcripts enriched in synonymous codons that are rarely used in the genome[16,19,39–41]. Such distinct regulation of mRNAs with rare codons mirrors other unique aspects of mRNA regulation in neural cells, including splicing[42–44], RNA modifications[45,46], and polyadenylation[47,48]. *Drosophila* larvae have a well-characterized neural stem cell lineage[5,49–52], making the fly larval brain a well-suited system to explore dynamic codon usage regulation during neural differentiation.

In this study, we use genetic tools, quantitative microscopy, and RNA sequencing in the developing *Drosophila* brain to reveal an upregulation of rare-codon-enriched mRNA and protein expression during neural differentiation. Neurons, but not neural progenitors, upregulate protein expression from a rare-codon-enriched reporter during differentiation. This shift in genetic code regulation is driven by increased rare-codon-enriched mRNA abundance and protein production in neurons compared to neural stem cells. From a candidate

[1]Department of Pharmacology & Cancer Biology, Duke University, Durham, NC, USA. [2]Duke Regeneration Center, Duke University, Durham, NC, USA. [3]Department of Biology, University of North Carolina, Chapel Hill, NC, USA. [4]Department of Biology, Duke University, Durham, NC, USA. ✉e-mail: don.fox@duke.edu

genetic screen, we identify the conserved CPEB protein Orb2 as a specific regulator of rare-codon-biased protein expression in neurons and in spermatids of the testis. These findings mirror Orb2's known roles in both neuronal function[53–60] and in male fertility[61–63]. Using RNA sequencing, we then reveal that Orb2-upregulated brain mRNAs with abundant Orb2 binding sites have a rare-codon bias. From these Orb2-upregulated mRNAs, we show that rare codons are critical for expression and social behavior function of the G protein-coupled receptor *mGluR*. Codon optimization of mGluR increases its expression, and this negatively impacts social behavior. We further show that Orb2 controls mRNA stability of mGluR, and also controls mRNA stability of a GFP reporter in a codon-dependent fashion in neurons. Our study identifies a molecular mechanism through which differentiation in a stem cell lineage shifts genetic code regulation to enable critical protein expression and function.

## Results

### Protein production from rare-codon-enriched genes in the *Drosophila* brain occurs during neuronal differentiation

We previously showed that multiple GFP reporters enriched in rare codons and driven by a ubiquitous (UBI) promoter express specifically in the brain and testis of *Drosophila melanogaster*[16]. Expression in these two tissues vastly exceeds expression in any other tissue. We also identified a reporter (*GFP-rare-brain*, hereafter *GFP-rare[b]*) with highly specific wandering third larval instar (WL3) brain expression (Fig. 1A, B, Fig. S1B, GFP-rare[b]). *GFP-rare[b]* contains 54% common codons at the 5' end of the transgene followed by 46% rare codons at the 3' end. This codon usage and clustering of rare codons at the 3' end is similar to our previous rare-codon reporters, but *GFP-rare[b]* uses a distinct nucleotide sequence (Supplementary Dataset 1). Given the robust and specific expression in the brain, we used *GFP-rare[b]* to uncover regulation of rare-codon-enriched transcript expression at the cellular level.

High magnification imaging reveals that GFP-rare[b] protein expression across the brain is non-uniform, suggesting that only specific cell types may express this rare-codon-enriched reporter (Fig. 1B). To identify codon-dependent expression differences in cell types in the larval brain, we compared GFP-rare[b] expression to a previously described reporter with the same promoter and genomic insertion location (see Methods). This reporter, here termed *GFP-common*, contains 100% common codons and is expressed ubiquitously throughout the larva[13,16] (Fig. 1A, C, Fig. S1A, GFP-common). GFP-common protein is expressed more uniformly in the brain (Fig. 1C), suggesting the cellular level difference in GFP-rare[b] brain expression is codon-dependent. We confirmed the codon dependency of this expression using other brain-enriched, rare-codon-biased reporters, and observe the same non-uniform fluorescence pattern seen with GFP-rare[b] (Fig. S1C, D). Therefore, only some cells in the brain can express GFP-rare[b].

To determine which brain cell types express GFP-rare[b], we used both antibodies and genetic markers specific for neural cell types. We focused on the well-characterized central brain neuroblast lineages. Cell types and lineage relationships common to all neuroblast lineages are schematized in Fig. 1D. First, we looked at the neuroblasts (NBs), which are the neural stem cells. Using an antibody for the NB-specific transcription factor Deadpan (Dpn)[64], we find that GFP-rare[b] is not expressed in NBs, while GFP-common is (Fig. 1E vs F). NBs divide asymmetrically to self-renew and produce another type of progenitor cell, ganglion mother cells (GMCs)[5,49–52,65,66]. Using *earmuff*-GAL4 to drive a membrane-targeted UAS-tdTomato protein specifically in GMCs[67], we find GFP-rare[b] protein is again not expressed, while GFP-common is expressed (Fig. 1G vs H). GMCs divide to produce post-mitotic progeny that differentiate into either neurons or glia. We used an antibody against Elav, an RNA binding protein, to mark neurons specifically[68]. GFP-rare[b] is expressed in Elav-positive neurons, but in a spatially distinct way. Neurons more distal from the NB do express

GFP-rare[b], while neurons near the NB do not (Fig. 1I, solid vs dotted line). We confirmed using MARCM clones (see Methods[69]) that these proximal neurons are more recently born and immature (Fig. S1E). GFP-common protein is expressed throughout all neurons in the central brain lineage (Fig. 1J). We also examined GFP-rare[b] in larval neurons and glia outside of the central brain. Most of these cell types, such as OK6-Gal4+ motor neurons and en-Gal4+ serotonergic neurons, express GFP-rare[b] (Fig. S1F, G). However, we did detect a small number of cell types that are negative for GFP-rare[b], notably Wrapper+ midline glia and Repo+ astrocyte-like glia (Fig. S1H, I). These results show that most differentiated larval brain cells uniquely upregulate GFP-rare[b] protein.

Our marker analysis suggests that, in the larval central brain NB lineage, the ability to express a rare-codon-enriched reporter is coupled to cell differentiation. To test this model, we genetically manipulated NB division and progeny differentiation. We did so by increasing the number of undifferentiated NBs in the brain at the expense of differentiated neurons and glia. Decreased expression of *numb*[70–72] or *prospero (pros)*[73,74] is known to increase the number of a prevalent type of NBs in the *Drosophila* brain known as the Type 1 NB. Similarly, decreased expression of *brain tumor (brat)*[65,66,70,75,76] increases the number of highly proliferative Type 2 NBs. We recapitulated these results by driving expression of UAS-RNAi constructs against the above-mentioned genes using the *inscuteable (insc)*-GAL4 driver, which is expressed throughout all NBs and their progeny[77]. We find significant increases in the number of Dpn+ Type 2 NBs in the *brat* RNAi background (Fig. 1K, L; Fig. S1K) and Dpn+ Type 1 NBs in the *numb* and *pros* RNAi backgrounds (Fig. S1J). If cellular differentiation is required to express GFP-rare[b], then GFP-rare[b] expression should decrease when the percentage of NBs increases. Indeed, RNAis that increase numbers of NBs or intermediate progenitors (and decrease numbers of differentiated cells) leads to smaller areas of GFP-rare[b] protein expression (Fig. 1M). These results show that cell differentiation within the NB lineages determines the ability to express a rare-codon-enriched reporter.

### Neurons but not neuron progenitors upregulate rare-codon-enriched mRNA expression

Regulation of protein expression from a rare-codon-enriched mRNA can occur at the level of RNA or protein (reviewed in ref. 78,79). We next examined if neurons uniquely upregulate *GFP-rare[b]* mRNA in addition to protein. Using quantitative microscopy (see Methods) and genetically encoded fluorescent markers (Fig. S2A, B), we first measured the amount of GFP-rare[b] protein at each step in the NB lineage. There is almost 10-fold more GFP-rare[b] protein by fluorescence in neurons than in either NBs or GMCs (Fig. 2A, B, Fig. S2C). There is only a twofold difference in GFP-common protein between neurons and NBs, with more protein in the NBs (Fig. 2D, E, Fig. S2D). Using western blot, we also quantitatively measured expression of GFP-rare[b] and GFP-common protein in both WT and NB-enriched *brat* RNAi brains. In agreement with our findings by microcopy in Fig. 1M, increasing the number of NBs significantly decreases the amount of GFP-rare[b] in brains (Fig. S2E). *brat* RNAi has no impact on GFP-common expression by western blot (Fig. S2E). Taken together, these data highlight a response in neurons which increases protein expression from a rare-codon-enriched reporter.

To examine mRNA level differences between NB lineage cells, we used single molecule inexpensive Fluorescent In Situ Hybridization (smiFISH, see Methods[80]). There is almost no *GFP-rare[b]* RNA in the NBs, similar to GFP-rare[b] protein (Fig. 2A–C). *GFP-rare[b]* RNA is detectable in GMCs, but neurons have a twofold higher level of GFP-rare[b] RNA compared to GMCs (Fig. 2C, GMCs vs. neurons). In contrast to the stepwise RNA level increases from progenitors to neurons seen with GFP-rare[b], GFP-common mRNA does not increase in neurons (~1.3-fold higher in NBs, Fig. 2D, F). We also measured total reporter mRNA in

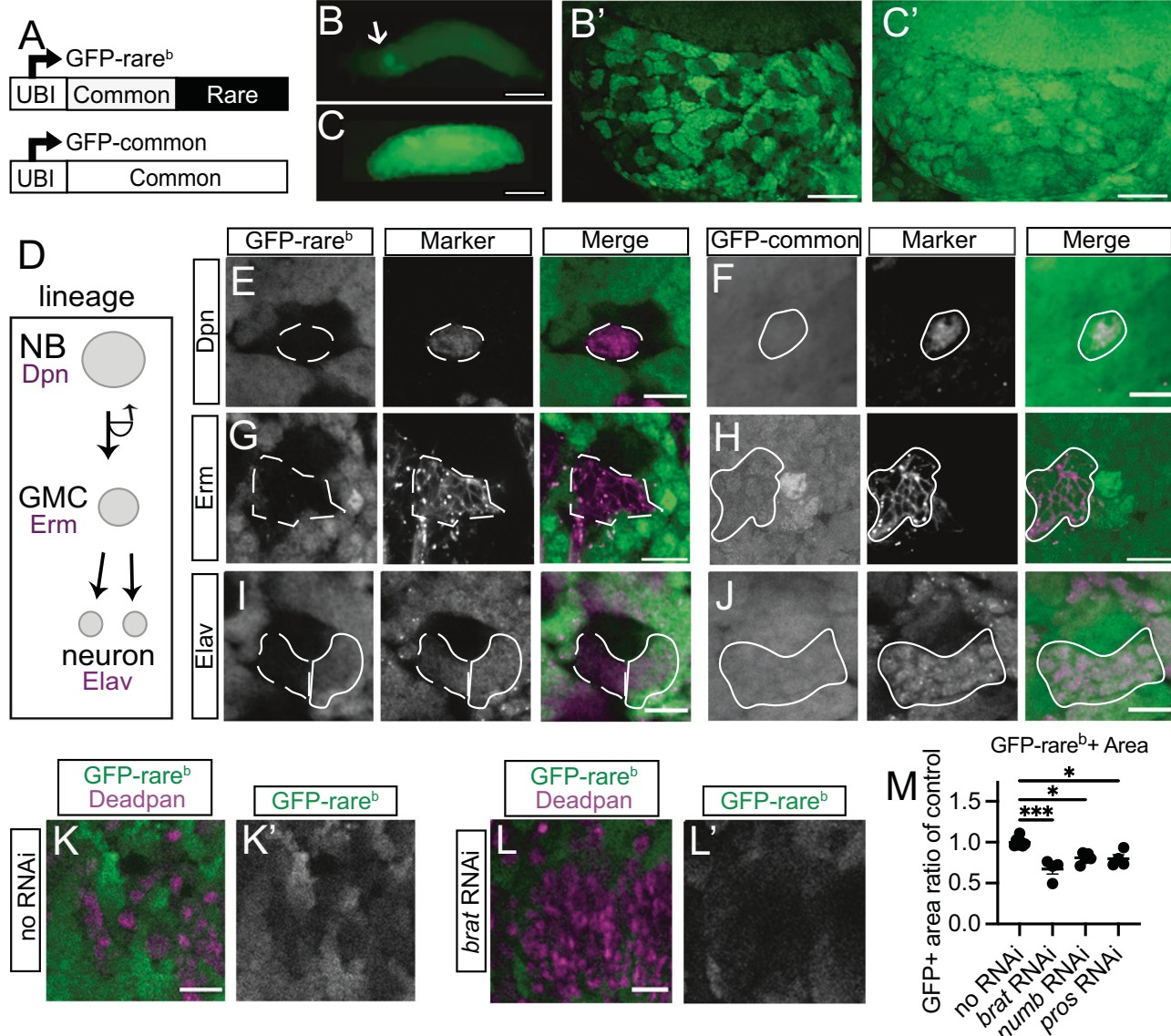

**Fig. 1 | Expression of a rare-codon-enriched reporter during differentiation of *Drosophila* neurons. A** Schematic of GFP-rare[b] and GFP-common reporters. 5′ is to the left. **B** Expression of GFP-rare[b] in whole WL3 larvae, scale bar is 1 mm. Arrow labels the brain. **B′** Expression of GFP-rare[b] in one lobe of a WL3 brain, scale bar is 20 μm. **C-C′** Expression of GFP-common in WL3 larvae as in **B**, **B′**. **D** Schematic of neuroblast lineage in *Drosophila* larval brain, showing markers of neuroblasts (NBs), ganglion mother cells (GMCs) and neurons in magenta. **E–J** Representative images of WL3 brain expressing GFP (green), either GFP-rare[b] (**E**, **G**, **I**) or GFP-common (**F**, **H**, **J**). Brains are co-labeled with cell type markers (magenta). Deadpan (Dpn) marks NBs (**E**, **F**), *earmuff (erm)* GAL4 drives membrane-targeted tdTomato in GMCs (**G**, **H**), and Elav marks neurons (**I**, **J**). Solid outline shows cells expressing marker and GFP protein. Scale bar is 10 μm. **E**, **G**, **I** GFP-rare[b] reporter, dashed outline shows cells expressing marker that do not express GFP protein. **K-L′:** Representative images of WL3 brains expressing GFP-rare[b] (green on left, white on right) co-labeled with Dpn (magenta on left) to mark NBs. Scale bar is 20 μm. **K-K′** GFP-rare[b] reporter brain. **L-L′** GFP-rare[b] reporter brain with *inscuteable (insc)* GAL4 driving UAS-*brat* RNAi in the NB lineage. **M** Quantification of area of central brain covered by GFP-rare[b] expressing cells in areas affected by each RNAi. Each data point = 1 animal, mean and SEM plotted. Two biological replicates conducted, n for each genotype are as follows: no RNAi = 7, *brat* RNAi = 5, *numb* RNAi = 5, *pros* RNAi = 5. All comparisons by one-way ANOVA followed by Dunnett's multiple comparisons tests comparing no RNAi to: *brat* RNAi, $p = 0.0008$; *numb* RNAi, $p = 0.0378$, *pros* RNAi, $p = 0.0412$. Source data are provided as a Source Data file.

brains using quantitative Real-Time PCR (qRT-PCR) and found there is significantly less *GFP-rare[b]* mRNA compared to *GFP-common* mRNA (Fig. S2F). We then calculated the ratio of reporter mRNA and protein in NBs and neurons by normalizing the amount of *GFP-rare[b]* RNA in each cell type to the amount of GFP-common reporter RNA, then dividing by the amount of GFP-rare[b] protein (see Methods). This shows that neurons have roughly 5-fold higher GFP-rare[b] protein output per mRNA than neuroblasts (Fig. 2G). Taken together, our RNA and protein measurements suggest that neurons, but not neural progenitors, have a unique ability to preferentially maintain levels of and produce protein from rare-codon-enriched mRNAs. As GFP-common and GFP-rare[b]

have the same genomic location and promoter, the 10-fold GFP-rare[b] protein and twofold higher mRNA difference in cell type expression is primarily codon-dependent as opposed to reflecting cell type differences in transgene expression strength. Indeed, when comparing protein and mRNA levels in animals heterozygous and homozygous for *GFP-rare[b]*, the ratio correlates closely with gene dosage in each cell type (Fig. S2G, H).

Our study of *GFP-rare[b]* expression suggests that RNA levels in NBs and neurons are predictive of protein production from a rare-codon-enriched transcript. To determine if endogenously expressed mRNAs behave similarly to our reporter, we analyzed codon usage of

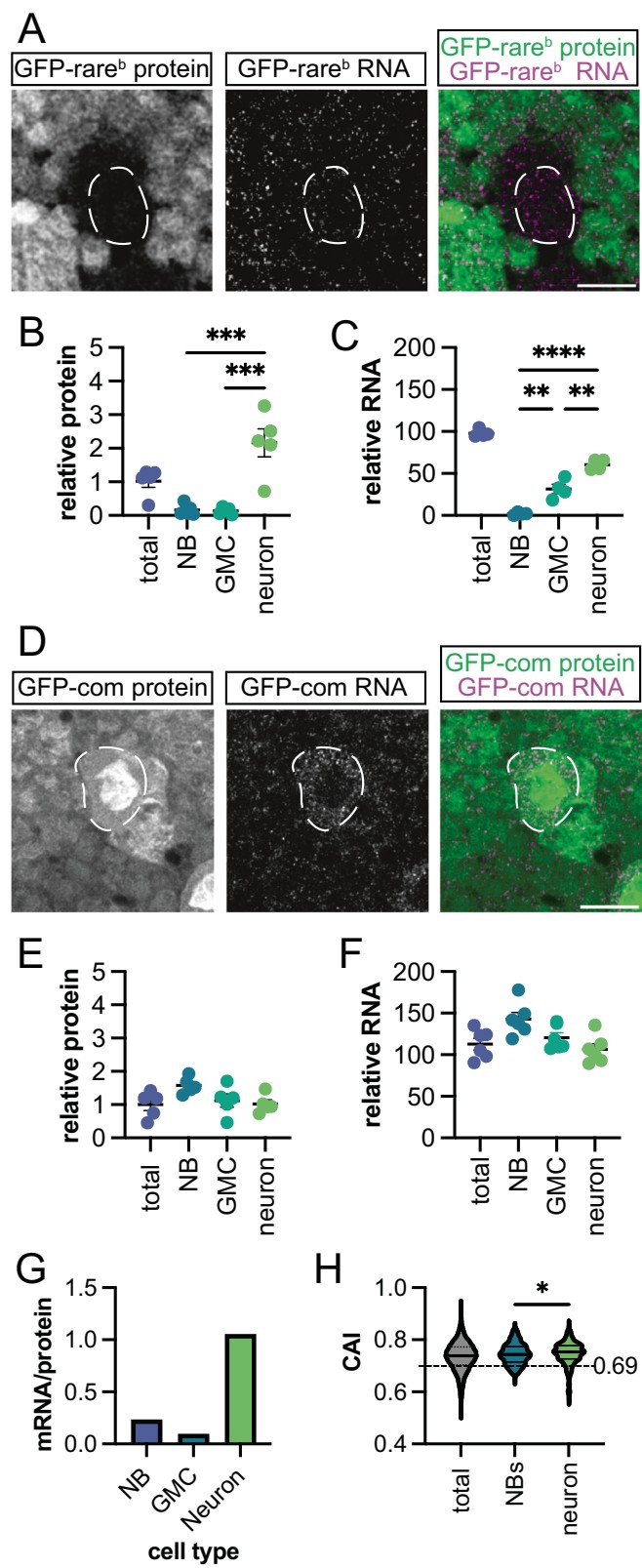

**Fig. 2 | Neurons but not neuroblasts upregulate rare-codon-enriched expression. A** Images of GFP-rare[b] protein (green in merged image) and mRNA (magenta in merged image) in WL3 brains. Scale bar 10 µm. Dashed outline indicates NB. **B** GFP-rare[b] protein expression relative to $w^{1118}$ controls in indicated cell types in WL3 brains (see Methods). Two biological replicates. One-way ANOVA performed followed by Tukey's multiple comparisons test, $p = 0.0002$ for both. **C** Quantification of GFP-rare[b] RNA expression relative to $w^{1118}$ controls in indicated cell types in WL3 brain (see Methods). Two biological replicates. One-way ANOVA performed followed by Tukey's multiple comparisons test. NB vs GMC, $p = 0.001$; NB vs neurons, $p < 0.0001$; GMC vs neurons, $p < 0.0011$. **D** Representative images of GFP-common protein (green in merged image) and mRNA (magenta in merged image) in WL3 brains. Scale bar 10 µm. Dashed outline indicates NB. **E** Quantification of GFP-common protein expression in indicated cell types relative to $w^{1118}$ controls in WL3 brains. Two biological replicates. No significant differences detected by one-way ANOVA. **F** Quantification of GFP-common RNA expression in indicated cell types relative to $w^{1118}$ controls in WL3 brain (see Methods). Two biological replicates. No significant differences detected by one-way ANOVA. **G** mRNA/protein ratio of GFP-rare[b] in each cell type. Relative GFP-rare[b] mRNA and protein from **E** and **F** are plotted as a ratio of mRNA to protein. **H** Analysis of codon usage in NBs and neurons from single-cell RNA sequencing data retrieved from Yang et al. 2016 and Berger et al. 2012 (see Methods); highly expressed genes were defined as twofold enriched cell type of interest in both datasets. Expressed genes in each cell type are represented in a violin plot of CAI distribution. "Total" is a plot of all mRNAs in the dataset. Dashed line indicates CAI of GFP-rare[b]. A Kolgomorov–Smirnov test was performed to compare the distributions between NBs and neurons (*: $p = 0.0173$). Source data for **B**, **C**, **E**–**H** are provided as separate Source Data files. Mean ± SEM plotted on each graph. Precise $N$ values provided in source data file for **B**, **C**, **E**, **F**.

scores indicate an enrichment of common codons, with a score of 1 indicating that the most common codon is used in every position. This analysis reveals a significant difference in the distribution of CAIs for mRNAs between NBs versus neurons (Fig. 2H). Neurons but not NBs exhibit a prominent tail of low CAI (rare-codon-enriched) genes. These data mirror our findings with our *GFP-rare[b]* reporter. Using both a pair of codon-biased reporters and two published single-cell RNA sequencing datasets, we have discovered a unique upregulation of rare-codon-enriched mRNA and protein expression upon differentiation of NBs into neurons.

**A candidate screen identifies the conserved CPEB protein Orb2 as a regulator of codon-dependent expression**

We next used our reporter system in a targeted genetic screen to identify upstream regulators of codon-dependent expression during neuronal differentiation. We used available single-cell RNA sequencing data[81,82] to generate a list of candidate regulators. Given our finding of rare-codon expression differences between NBs and neurons, we reasoned that candidate upstream regulators of these differences have differential expression between NBs and neurons. Candidates were further selected based on a known function in RNA biology. We reasoned that rare-codon expression could be achieved by regulating stability of the mRNA through binding, splicing, or modification. Conversely, rare-codon expression could be impacted by translation, ribosome regulation, or tRNA modification. We chose 21 candidates enriched in NBs, and 24 candidates enriched in neurons (schematized in Fig. 3A). We knocked down each candidate with RNAi expressed by *insc*-GAL4. For five of these genes (*CG9107, dim1, futsch, lsm3, me31b*), knockdown by RNAi in the brain was lethal. For the remaining 40 genes, we measured the fluorescence intensity of GFP-rare[b] across one lobe of the central brain to assess regulation of rare-codon-dependent expression. This primary screen revealed that knockdown of five genes (*elp1, CG2021, not1, ewg* and *orb2*) significantly decreases GFP-rare[b] expression (Fig. 3B, C, Supplementary Dataset 2). As a secondary screen, we repeated the RNAi knockdown with *insc*-GAL4 in the background of the GFP-common reporter to determine if the effects of the knockdown are specific to codon content (Fig. 3D). Only

published single-cell RNA sequencing. Using previously published data[81,82], we analyzed the codon usage of mRNAs that are highly and specifically expressed in either NBs or neurons (see Methods). As a metric of codon usage, we used the codon adaptation index (CAI)[9]. CAI determines the likelihood that the set of codons making up a transcript will appear with that specific frequency, based on the frequency of appearance for all codons in the genome of the organism. Higher CAI

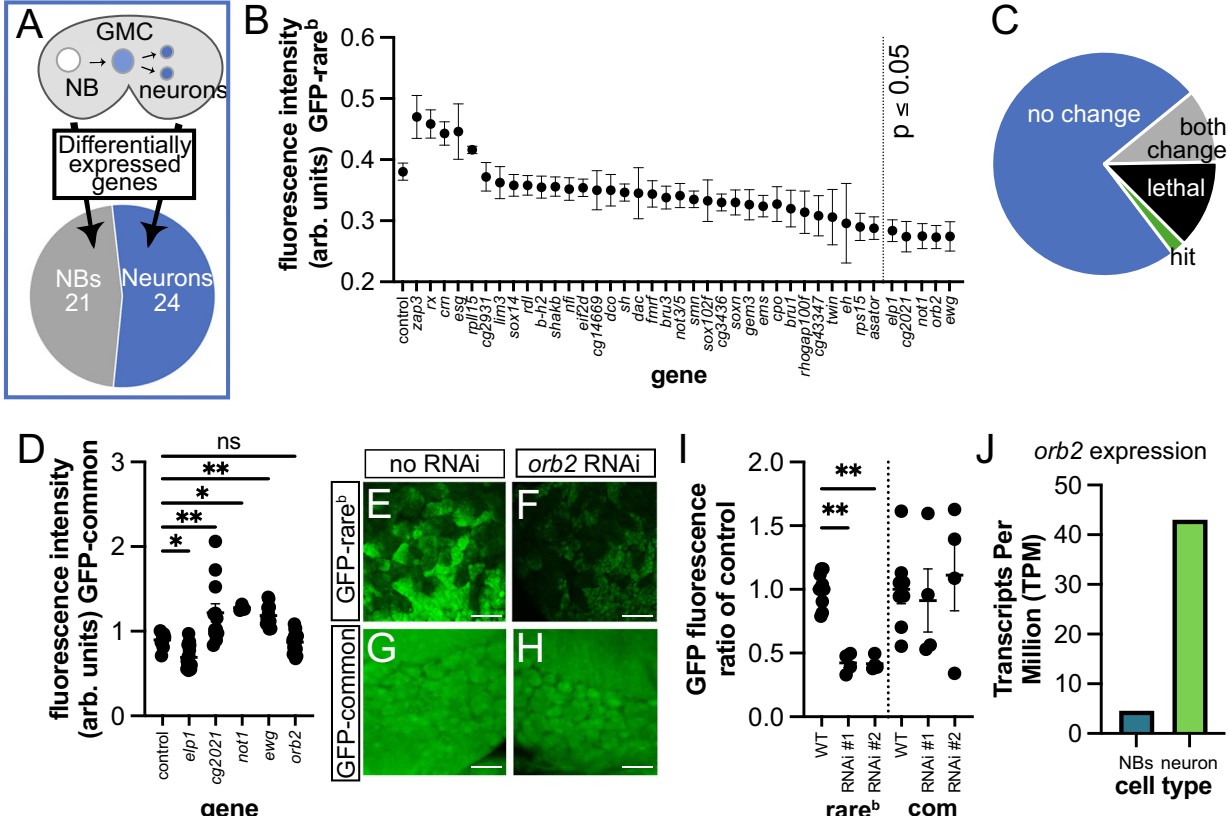

**Fig. 3 | A targeted RNAi screen reveals Orb2 as a codon-specific regulator in the brain. A** Schematic indicating how screen targets were chosen. 21 genes were NB-enriched, and 24 genes were neuron-enriched. **B** Quantification of primary screen results, with average GFP-rare[b] fluorescence intensity values (arb. units=arbitrary units) in target gene deficient brains (RNAi driven by *insc*-GAL4) ordered from highest to lowest mean. Ordinary one-way ANOVA compares all values to control (expressing GFP-rare[b] and no RNAi) WL3 brain fluorescence value followed by Dunnett's test for multiple comparisons. Knockdown of *elp1, cg2021, not1, ewg* and *orb2* significantly decreased average GFP-rare[b] WL3 brain fluorescence relative to the control. *P* values for significant comparisons with control as follows: *cg2021: p* value = 0.0437; *not1: p* value = 0.0203; *elp1: p* value = 0.0139; *ewg: p* value = 0.0018; *orb2: p* value = 0.0015. Mean ± SEM plotted. **C** Graphical demonstration of primary and secondary screen results. *N* = 35 RNAis with no change in GFP-rare[b] fluorescence, *N* = 6 RNAis were lethal by larval stage, *N* = 5 RNAis significantly changed fluorescence of both GFP-rare[b] and GFP=common, and *N* = 1 RNAi, (*orb2*), significantly changed only GFP-rare[b] fluorescence (="hit").

**D** Quantification of secondary screen results. Knockdown of primary screen hits driven by *insc*-GAL4 results in significant changes to GFP-common fluorescence in all but *orb2* RNAi knockdown. Ordinary one-way ANOVA followed by Dunnett's test for multiple comparisons. Compared to control: *elp1 p* = 0.034; *cg2021, p* = 0.0011; *not1, p* = 0.0142; *ewg, p* = 0.0040. Each data point = 1 animal. Mean ± SEM shown. Three biological replicates. **E–H** Representative images of control (*white* (*w*) RNAi) (**E, G**) and *orb2* RNAi (**F, H**) WL3 brains with GFP-rare[b] (**E, F**) or -common (**G, H**) fluorescence (green). **I** Quantification of fluorescence intensity change in *insc*-GAL4>*w* RNAi (WT) or *orb2* RNAi WL3 brains compared to average fluorescence in WT, expressing either GFP-rare[b] or GFP-common. Two separate *orb2* RNAi constructs were tested (see Supplemental Dataset 4 for details). Mean ± SEM plotted, two biological replicates. Ordinary one-way ANOVA followed by Šídák's multiple comparisons test. *P* value for GFP-rare[b] vs RNAi#1 = 0.0096; GFP-rare[b] vs RNAi#2 = 0.0087. **J** *orb2* transcript expression in NBs and neurons from Yang et al. 2016[82]. Precise *n* values in source data files for **B**, **D** and **I**.

knockdown of *orb2*, an RNA binding protein, significantly decreases expression of GFP-rare[b] without changing GFP-common expression (Fig. 3D–H). The codon-dependent effect on GFP-rare[b] versus GFP-common expression was repeated with a second *orb2* RNAi line (Fig. 3I) and with an *orb2* deletion mutant (Fig. S3A). By analyzing single-cell RNA sequencing data[82], we observe a 10-fold higher mRNA expression of *orb2* in neurons vs NBs. (Fig. 3J). This differential expression suggests that neuronal upregulation of *orb2* expression may underlie the robust expression of rare-codon-enriched mRNA and protein in neurons.

To determine how Orb2 impacts codon-dependent mRNA and protein within the NB lineage, we used quantitative microscopy to simultaneously measure reporter protein and RNA expression (Fig. 4A–D). First, we compared GFP-rare[b] protein expression between brains of *orb2* RNAi (driven by *insc*-Gal4) and a negative control (*insc*-GAL4 driving *white* RNAi). Except where specifically noted, we use *insc*-Gal4 driving *orb2* or *white* RNAi for all subsequent *orb2* vs. control comparisons in this study. RNAi against *orb2* does not impact GFP-rare[b]

protein expression in NBs or GMCs (Fig. 4C). However, in neurons, there is a 50% decrease in GFP-rare[b] protein expression in *orb2* RNAi relative to *white* RNAi. This suggests that *orb2* positively regulates GFP-rare[b] expression specifically in neurons. We observe similar results by driving *orb2* RNAi with *elav*-GAL4, which expresses only in neurons[68] (Fig. S3B–E). We do not observe any difference in GFP-common protein expression in *orb2* RNAi brains (Fig. S3F). Importantly, *orb2* loss does not alter the number of NBs or Elav+ cells in brains, suggesting that changes in GFP-rare[b] protein expression are not due to secondary impacts on the NB lineage (Fig. S3G, H). As *orb2* is an RNA binding protein, we next compared GFP-rare[b] mRNA levels in *orb2* RNAi and *white* RNAi WL3 brains. Loss of *orb2* does not impact GFP-rare[b] mRNA levels in NBs or in GMCs. However, there is a significant decrease in GFP-rare[b] RNA expression in the total brain in *orb2*-RNAi animals, seemingly driven by the decrease in mRNA abundance in neurons, as measured by quantitative microscopy and qRT-PCR (Fig. 4D, Fig. S3I). There is no change in GFP-common mRNA levels in *orb2* RNAi brains by qRT-PCR comparing the amount of reporter mRNA in total brains from

control (*w* RNAi) animals to *orb2* RNAi animals (Fig. S3I). Similarly, by quantitative microscopy with smFISH, there is no difference in GFP-common mRNA in the different cell types in the neuroblast lineage (Fig. S3J). These data suggest that the RNA binding protein Orb2 acts specifically in neurons to increase abundance of GFP-rare[b] mRNA and thereby increase protein production from the mRNA.

Orb2 is connected to mRNA stability[55,59–61,83–85]. We therefore examined if Orb2 impacts mRNA stability of GFP reporters in a codon-

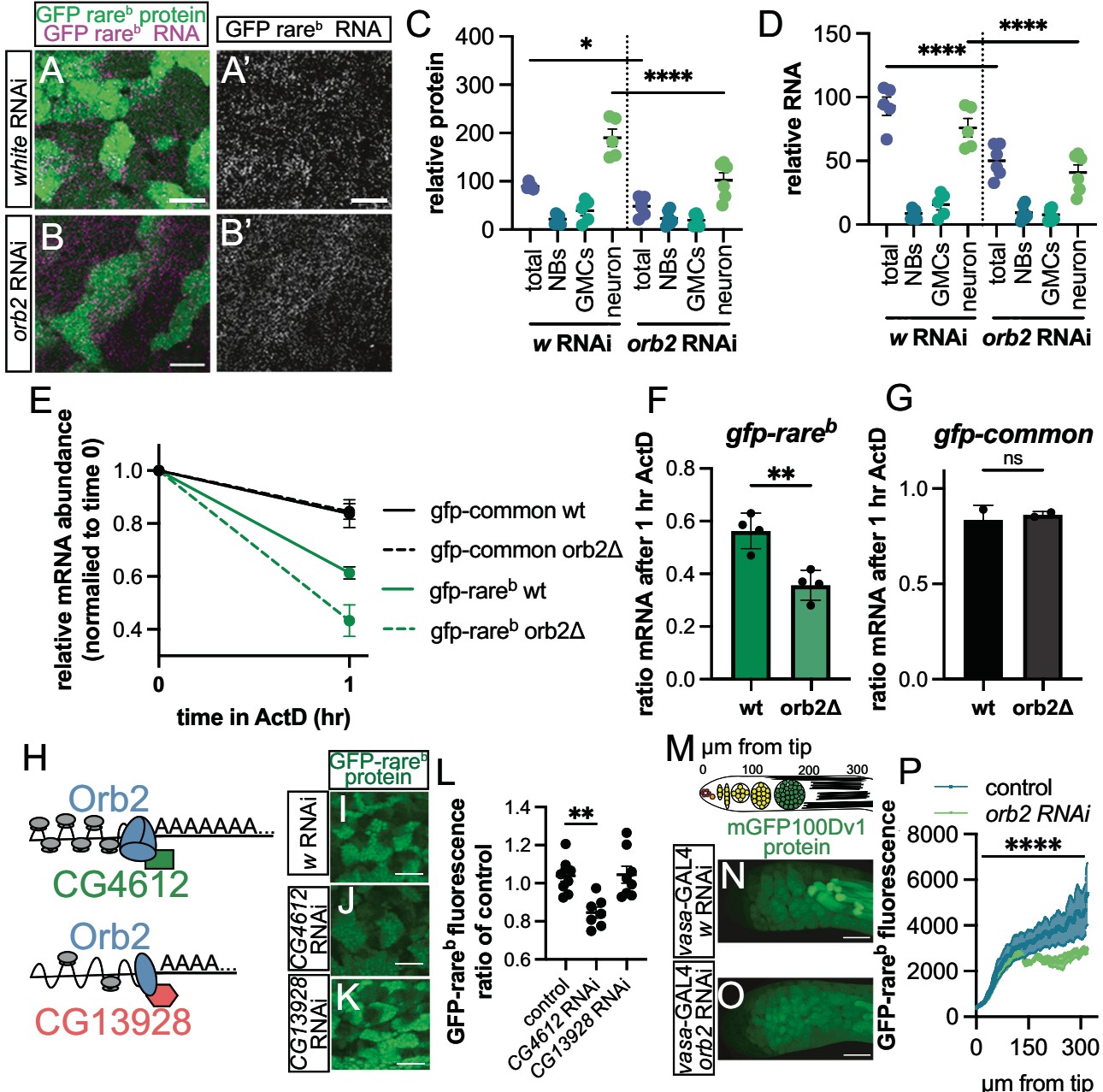

**Fig. 4 | Molecular and cellular evidence of Orb2 as a positive regulator of rare-codon-enriched expression. A-B'** Images of GFP-rare[b] protein (green on left) and mRNA (magenta on left, white on right) in WL3 brains, *insc*-GAL4>*w* RNAi (A-A') or *insc*-GAL4>*orb2* RNAi (B-B'). Scale bar 10 μm. **C** GFP-rare[b] protein decreases with *orb2* RNAi in total brains and neurons. Two biological replicates, one-way ANOVA followed by Šídák's multiple comparison test. *w* RNAi vs *orb2* RNAi total *p* value = 0.0268. *w* RNAi vs *orb2* RNAi neurons *p* value < 0.0001. **D** GFP-rare[b] RNA decreases with *orb2* RNAi in total brains and neurons. Two biological replicates, *w* RNAi vs *orb2* RNAi total and neurons *p* < 0.0001 by one-way ANOVA followed by Šídák's multiple comparison test. **E** Change in mRNA abundance over one hour of ActD treatment. Mean ± SEM, 20 brains in each condition, 3 replicates for GFP-rare[b] and 2 replicates for GFP-common. **F, G** Plot of ratio of mRNA over 1 hr of ActD treatment with or without *orb2*. **F** GFP-rare[b]. Two-tailed *t* test, *p* = 0.0035. **G** GFP-common. Two-tailed *t* test, *p* not significant (ns). **H** Schematic of Orb2, CG4612, and CG13928 influencing mRNA polyadenylation and translation. Modified from Khan et al. 2015. **I–L** Images of GFP-rare[b] protein in WL3 brains with *insc*-GAL4> *w* RNAi (**I**), *CG4612* RNAi (**J**), *CG13928* RNAi (**K**). **L** GFP-rare[b] protein expression in total brain for *w* RNAi, *CG4612* RNAi, and *CG13928* RNAi normalized to average of all *w* RNAi brains. Two biological replicates. One-way ANOVA followed by Dunnett's multiple comparisons test, control vs *CG4612* RNAi *p* = 0.0018. **M** Schematic of testes shown in **N** and **O** and measured in **P**. μm from tip corresponds to X axis of **P**. Red = hub, orange = germline stem cell, yellow = spermatogonia, green = spermatocytes, and black = spermatids. **N, O.** Images of testes GFP fluorescence from mGFP100Dv1 with *vasa*-GAL4> (**N**) *w* RNAi or (**O**) *orb2* RNAi. Scale bar 50 μm. **P** mGFP100DV1 in testes with *vasa*-GAL4>*w* RNAi or *orb2* RNAi. Line profiles of GFP-rare[b] fluorescence from hub to spermatid (300 μm). *N* = 9, mean ± SEM, three biological replicates. Welch's two tailed *t* test comparing linear regressions *p* < 0.0001. Precise *n* values for **C**, **D**, and **L** provided in source data file.

dependent manner. We first examined if rare codons decrease reporter mRNA stability. We used Actinomycin D (ActD) to stop transcription[86–88] and measured the abundance of *GFP-rare*[b] mRNA over time. Compared to *GFP-common, GFP-rare*[b] mRNA abundance decreases much more rapidly after an hour of ActD treatment, indicating it is less stable overall (Fig. 4E). We next examined *GFP-common* and *GFP-rare*[b] mRNA stability in *orb2* brains. Loss of *orb2* makes *GFPrare*[b] mRNA even less stable (Fig. 4E, F) but has no effect on *GFP-common* mRNA (Fig. 4E, G). These results highlight a mechanistic role for Orb2 in codon-dependent mRNA stability in the brain.

Next, we used genetic tools to further dissect the mechanism through which Orb2 is affecting GFP-rare[b] protein and RNA expression in the *Drosophila* brain. Previous work has shown that Orb2 can both activate and repress polyadenylation on target transcripts, leading to increased or decreased mRNA stability respectively[55,59,83–85]. The switch between stabilizer and destabilizer is determined in part by which proteins are recruited by Orb2 to the target transcript (schematized in Fig. 4G). Such regulation is important in the brain to regulate mating behavior[55]. Our data suggests that Orb2 enables higher protein production from rare-codon-enriched mRNAs (Figs. 3I, 4C–G). Given this, we predicted that knockdown of the Orb2 co-factor *CG4162* (required for activation, Fig. 4H) would similarly be required for rare-codon expression. Indeed, we find a significant decrease of GFP-rare[b] protein expression with *insc-Gal4*-driven *CG4612* knockdown (Fig. 4I, J, L). In contrast, *insc-Gal4*-driven depletion of the Orb2 co-factor *CG13928* (required for repression, Fig. 4H) has no impact on GFP-rare[b] protein expression (Fig. 4I, K, L). These results were duplicated with multiple RNAis for each Orb2 co-factor (Fig. S3K). Our findings with a rare-codon reporter suggest a mechanism whereby Orb2 and its activation co-factor CG4612 enable rare-codon expression in neurons.

In addition to in the brain, Orb2 also increases stability of testes-specific mRNAs with a concomitant increase in protein production from these mRNAs in the *Drosophila* testes[61,62]. Given that the testes and brain are the two specific tissues where we previously identified rare-codon-dependent expression[16], we next investigated the role of Orb2 in testes. To do this, we knocked down *orb2* with the germ cell driver *vasa*-Gal4 while co-expressing the rare-codon-enriched *mGFP100DV1* transgene, which expresses in both testes and brain[16]. Previously, we described the pattern of mGFP100DV1 protein expression in the testes. The tip of the testes (near the hub) does not express the *ubiquitin* promoter[16], as revealed by the absence of GFP-common (Fig. S3L, M left-most region). However, there is expression of both GFP-common and mGFP100DV1 throughout the spermatogonia, spermatocytes, and spermatids (stages schematized in Fig. 4M, images in Fig. 4N, O, Fig. S3L, M). *vasa* Gal4-driven *orb2* RNAi testes show less mGFP100DV1 protein expression than *vasa* Gal4-driven *white* RNAi controls (Fig. 4N–P). We measured mGFP100Dv1 protein expression from the hub to 300 microns proximal (schematized in Fig. 4M) and found the decrease in protein expression driven by *orb2* RNAi appears at the spermatocyte and spermatid stages (Fig. 4O, P). These stages are also where Orb2 has been previously described to activate translation of target mRNAs[61–63]. As in the brain, we do not observe an impact on GFP-common expression in *orb2* depleted testes (Fig. S3L–N). These data suggest that Orb2, which has specific roles in neuronal control of mating behavior and in spermatogenesis in testes[54,59,61,63], positively regulates rare-codon-dependent expression in neurons and spermatids.

## Orb2-regulated brain mRNAs have a codon-biased distribution of annotated Orb2 binding sites

Our results implicate Orb2 in recognizing rare-codon-enriched mRNAs and promoting their translation in neurons (and likely also in spermatids). Orb2 is known to bind a U-rich consensus sequence located within the 3′ UTR[56,60,84,85]. It has also been proposed that RNA binding proteins known to bind the 3′ UTR can also regulate targets through

the coding sequence (CDS)[89–91]. As all of our reporter transcripts contain the same 3′ UTR[16], it is unlikely that our finding of Orb2 as a codon-dependent regulator is driven by the 3′UTR. We therefore examined the number of Orb2 binding sites in the CDS. We used in silico analysis to examine the relationship between codon usage and the presence of Orb2 binding motifs in the reporter transcripts GFP-rare[b], GFP-common, and several other rare-codon reporters described previously[16]. Using the program RBPmap[92], we measured predicted Orb2 binding along each codon-modified transgene sequence.

There is a clear inverse correlation among our reporters between CAI and the number of annotated Orb2 binding sites (Fig. 5A). GFP-rare[b] has 6 annotated binding sites, while GFP-common has 0 (Fig. 5A, Fig. S4A, B). Within the GFP-rare[b] transcript, the putative Orb2 binding sites are in the rare-codon-enriched region of the transcript and near the 3′ end of the transcript, where Orb2 is modeled to bind[55,83] (Fig. S4A, B). In fact, depending on the reading frame, we find that rare codons and annotated Orb2 binding sites can converge (Fig. 5B). At CAI values below 0.6, Orb2 binding sites are highly abundant but the high level of rare codons prevents any reporter expression[16] (Fig. 5A). Further, the presence of abundant Orb2 binding sites alone does not appear to promote testis and brain-specific reporter expression, as the reporter GFP30D has a nearly identical CAI as GFP-rare[b] yet is ubiquitously expressed (Fig. 5A). This indicates there is a sweet spot where both rare-codon abundance and Orb2 regulatory action converge and allow tissue-specific protein expression from rare-codon-enriched genes.

These data suggest that both rare-codon enrichment and Orb2 binding motifs are important to determine regulation by Orb2 in the brain. However, it remained possible that Orb2 binding sites alone are sufficient to promote reporter expression in any cell type with sufficient Orb2 levels. To test this model, we used *insc-Gal4* to experimentally upregulate the Orb2 isoforms Orb2A and Orb2B individually and together throughout the neuroblast lineage[54,55,57–59]. We examined NBs, given that these stem cells normally have lower *orb2* mRNA levels (Fig. 3J). Neither Orb2 isoform alone nor both of them together is sufficient to promote GFP-rare[b] expression when upregulated (Fig. 5C), despite abundant Orb2 sites in this GFP reporter (Fig. 5A). Therefore, we conclude that in neurons but not NBs, Orb2 levels, Orb2 binding sites, and rare codons converge to promote rare-codon expression.

We next sought to identify endogenously expressed transcripts that are regulated by Orb2 in the brain. In doing so, we could then identify those Orb2-regulated transcripts with abundant rare codons and Orb2 binding sites. We conducted RNA sequencing to compare mRNA expression in control (*white* RNAi) and *orb2* RNAi WL3 brains. We found 1048 genes that are significantly decreased in expression when *orb2* is depleted (representing genes that are normally upregulated by Orb2) and 704 genes that are significantly increased in expression when *orb2* is depleted (representing genes that are normally downregulated by Orb2) (Fig. 5D, Supplementary Dataset 3). Among these differentially expressed genes, we conducted a similar correlation analysis between CAI and the frequency of Orb2 binding sites as we did for our GFP reporters. We again measured the number of potential Orb2 binding sites across the 3′ quartile of the CDS and the 3′ UTR of each transcript, focusing on mRNAs that are both differentially regulated by Orb2 and contain at least one annotated Orb2 binding site. Given a large number of Orb2-regulated genes, we split the genes into three groups by CAI: common-codon-enriched, middle-range codon usage, and rare-codon-enriched (see Methods). We first analyzed genes that, like *GFP-rare*[b], are upregulated by Orb2 (Fig. 5E, analysis without splitting by CAI in Fig. S4C). While there is no correlation between CAI and Orb2 binding site density for common-codon-enriched or middle-range codon usage Orb2-regulated genes, those with rare-codon enrichment show an inverse correlation, with more binding sites at lower CAI. These results suggest that there is a rare-codon bias to those brain mRNAs that are both positively regulated by

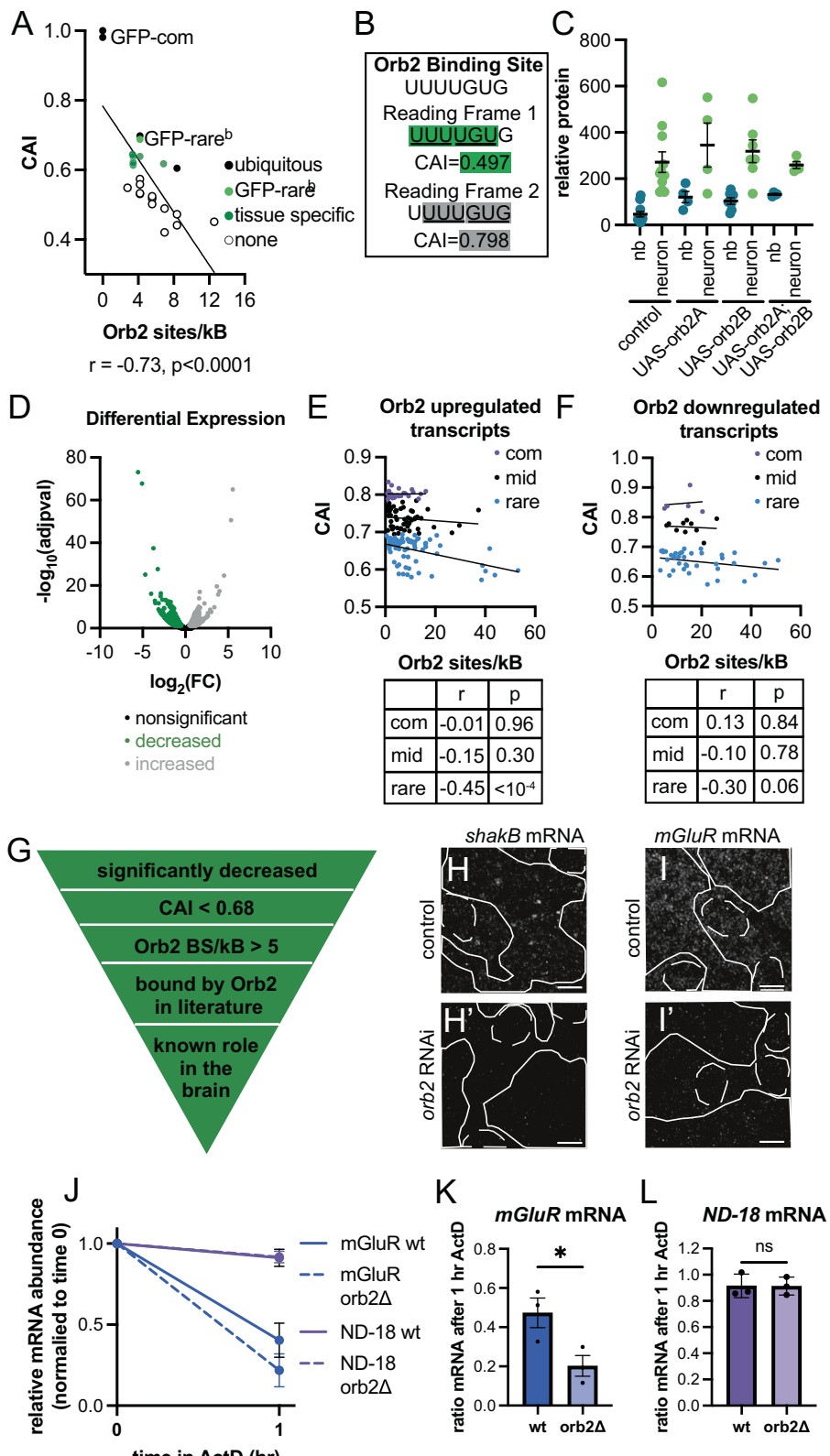

Orb2 and enriched in annotated Orb2 binding sites. In contrast to our results with mRNAs that are positively regulated by Orb2, we observe no correlation between CAI grouping and Orb2 binding site density for brain mRNAs that are downregulated by Orb2 (Fig. 5F, analysis without splitting by CAI in Fig. S4D). These data highlight a convergence between rare-codon enrichment and abundant annotated Orb2 binding sites in brain mRNAs that are positively regulated by Orb2.

Based on the relationship between rare codons and Orb2 binding sites in genes positively regulated by Orb2, we selected candidate genes for further analysis. We filtered our list of mRNAs that are positively regulated by Orb2 (1084 mRNAs) by identifying those with a CAI less than 0.68 (100 mRNAs), a value of more than 9 annotated binding sites per kilobase (42 genes), presence in previously published datasets of Orb2 binding mRNAs[60,85,90] (34 mRNAs), and a known role

**Fig. 5 | Orb2 increases abundance of endogenous rare-codon-enriched mRNAs with annotated Orb2 binding sites in the *Drosophila* brain. A** Scatterplot of annotated Orb2 binding sites and CAI measurements for codon-altered GFP transgenes[16] (see Methods). Linear regression reports *p* value < 0.0001, *n* = 28. Labeled are GFP-common and GFP-rare[b] (in bright green). Points are colored by level of expression in *Drosophila* larvae[16]: black is ubiquitous, green is tissue-specific, white is none. **B** Reading frame of Orb2 binding site determines CAI. Shifting the reading frame of Orb2 binding site[55,60] can lead to enrichment of rare (green) or common codons (gray). **C** Overexpressing *orb2* isoforms does not enable GFP-rare[b] expression in NBs. Control (*w* RNAi), UAS-orb2A and/or UAS-orb2B driven by *insc*-GAL4. Non-significant between control and all genotypes by One-way ANOVA. Mean ± SEM plotted, precise *n* values provided in source data file, 3 biological replicates. **D** Volcano plot of differential RNA expression profiles between *insc*-GAL4>*w* RNAi and *insc*-GAL4>*orb2* RNAi. Fold change was plotted as log$_2$(fold change) against adjusted *p* value, plotted as −log$_{10}$(adjusted *p* value). Genes with

significantly different fold change are plotted in green for negative and gray for positive. Statistics information provided in methods. **E, F** Scatter plots of annotated Orb2 binding sites of trnascripts with significantly changed fold change from RNA seq. See methods for details. Linear regression plotted, *r* and *p* value for each group shown in the table. **E** Transcripts upregulated by Orb2. **F** Transcripts downregulated by Orb2. **G** Candidate selection criteria for Orb2-regulated transcripts. **H, I′** *shakB* and *mGluR* mRNA smiFISH in WL3 brains. Dashed lines = NBs, solid lines = neurons. Scale bar 10 μm. One representative image of three animals from one of three biological replicates shown. **H-H′** *shakB* mRNA in *insc*-GAL4> *w*-RNAi (**H**) or *insc*-GAL4> *orb2*-RNAi (**H′**). **I-I′** *mGluR* mRNA in *insc*-GAL4> *w*-RNAi (**I**) or *insc*-GAL4>*orb2*-RNAi (**I′**). **J** Plot of change in mRNA abundance over one hour of ActD treatment. Mean ± SEM are plotted. 20 brains each condition, 2 replicates for *ND-18* and 3 replicates for *mGluR*. **K, L** Plot of mean ± SEM ratio of mRNA over timecourse of ActD treatment with or without *orb2*. **K** *mGluR* mRNA. Two-tailed *t* test, *p* = 0.043. **L** *ND-18* mRNA. Two-tailed *t* test, *p* value not significant (ns).

for the mRNA in the brain (16 mRNAs) (Fig. 5G). 10 example candidate mRNAs are highlighted in Fig. S3E. Using quantitative smiFISH, we validated our RNA-seq results by showing that *orb2* depletion decreases mRNA abundance for candidate genes *mGluR* and *shakB* (Fig. 5H, I). Both *mGluR* and *shakB* mRNAs are known to be direct Orb2 targets[56,85] with important neuronal functions. mGluR is a G protein-coupled receptor that functions in social activity and memory formation and influences adult brain development[91–93]. ShakB is an innexin that is necessary to form gap junctions in the *Drosophila* nervous system during development, which are maintained into adulthood and function in reflex responses[94,95]. These data suggest that Orb2 upregulates endogenous rare-codon-enriched mRNAs that are important for neuronal function.

We focused further on mGluR. *mGluR* has an abundance of rare codons (CAI = 0.585), placing it in the rarest 2% of the coding genome. The *mGluR* mRNA has 28 annotated Orb2 binding sites and was identified previously as an mRNA that directly interacts with Orb2[85,96]. Further, mGluR has a marked importance in neuronal development and function across evolution[97–99]. As our data with GFP reporters implicated mRNA stability as an important mechanism of codon-dependent Orb2 regulation, we again used actD treatment and quantitative RT-PCR to examine the role of Orb2 in *mGluR* mRNA stability. Relative to *ND-18*, a common-codon-enriched mRNA with a CAI of 0.862 shown previously to be highly stable in the nervous system[100], *mGluR* mRNA is less stable after an hour of ActD treatment (Fig. 5J). Similar to *GFP-rare[b]* reporter mRNA, *orb2* depletion makes *mGluR* mRNA even less stable (Fig. 5J, K). Depletion of *orb2* has no effect on *ND-18* mRNA stability, similar to *GFP-common* mRNA (Fig. 5L). These results suggest that Orb2 regulates the mRNA stability of *mGluR*, a rare-codon-enriched mRNA, in the brain. Our findings further support a mechanism whereby Orb2 regulates rare-codon-enriched mRNAs through mRNA stability control.

### Rare codons in *mGluR*, an Orb2-regulated mRNA, impact mating behavior

Our RNA-seq data suggests that Orb2 upregulates numerous rare-codon-enriched brain mRNAs, especially those with many annotated Orb2 binding sites. We next examined the role of both rare codons and Orb2 binding sites in *mGluR*. To study the role of annotated Orb2 binding sites and rare codons in mGluR function and expression, we generated three different transgenes: *mGluRendo*, *mGluRcom* and *mGluRcombs* (Fig. 6A–C). The *mGluRendo* transgene has the same coding sequence as the endogenous gene and contains 28 annotated Orb2 binding sites (Fig. 6A). The *mGluRcom* transgene modifies the *mGluR* nucleotide sequence to have the most common codons for each amino acid in the protein, and this sequence modification eliminates all annotated Orb2 binding sites (Fig. 6B). The *mGluRcombs* transgene has the most common codons for each amino acid at all positions in the protein except where most of the Orb2 binding

sites are, which have been changed back to the codons of the endogenous sequence, leading to 26 binding sites (Fig. 6C). All of these transgenes are expressed from the UBI promoter and tagged with a FLAG tag at the 5′ end and a 3XNBVHHO5 nanotag sequence at the 3′ end[101].

The correct level of mGluR expression is known to impact social activity in male flies[97,102]. We have previously shown that substituting rare codons in the *Drosophila Ras85D* mRNA can limit its expression and enable Ras signaling to be controlled by a distinct set of molecular regulators[13]. Similarly, we hypothesized that rare codons fine-tune mGluR expression, function, and regulation by Orb2. To test this hypothesis, we first examined if rare codons impact *mGluR* transgene brain expression and mating behavior function. We used quantitative microscopy to measure mGluR transgenic protein production. We examined both adult and larval stages (Fig. 6D, E) due to previously described roles for mGluR during development and adulthood[102]. We could not detect a codon-dependent difference in protein expression among the three *mGluR* transgenes in the adult brain (Fig. 6D). This result could reflect weaker expression by the UBI promoter in the adult brain, as we have observed previously[16]. In contrast, protein expression of both mGluRcom and mGluRcombs are significantly higher than mGluRendo in larval brains (Fig. 6E), consistent with a role for common codons in increasing protein expression during development. The quantitative fluorescent measurements were corroborated by quantitative western blots of whole brains from animals (Fig. S5A, B). These results suggest that rare codons limit expression of mGluR in the larval brain.

Previous studies found that mGluR activity during larval development is necessary for normal adult social activity in flies[102]. Similarly, embryonic expression of the mGluR ortholog is required for complete innervation of Purkinje cells in adult mice[103,104]. To assess if the codon-dependent expression of *mGluR* transgenes during larval development impacts the known role of mGluR in adult social behavior, we measured naïve courtship activity in adult males exposed to control females. As expected[105], control males spend a large percentage of their time courting (Fig. 6F). *mGluRendo* heterozygous males spend a similar amount of time courting as controls (Fig. 6F). However, *mGluRcom* heterozygous males spend a much smaller amount of time courting (Fig. 6F). This suggests that optimizing *mGluR* codons leads to higher mGluR protein during brain development and altered adult courtship behavior. *mGluRcombs* heterozygotes exhibit similar behavioral phenotypes as *mGluRcom* heterozygotes (Fig. 6F), suggesting that codon usage overrides the presence of sequences that may enable Orb2 regulation. To test this model further, we examined the ability of Orb2 to regulate transgenic mGluR. We measured the expression of each transgene in animals homozygous for an *orb2* deletion allele. *orb2* deletion significantly decreases expression of the *mGluRendo* transgene in the larval brain, suggesting that Orb2 can regulate this transgenic mGluR. In contrast, we observe no change in expression of

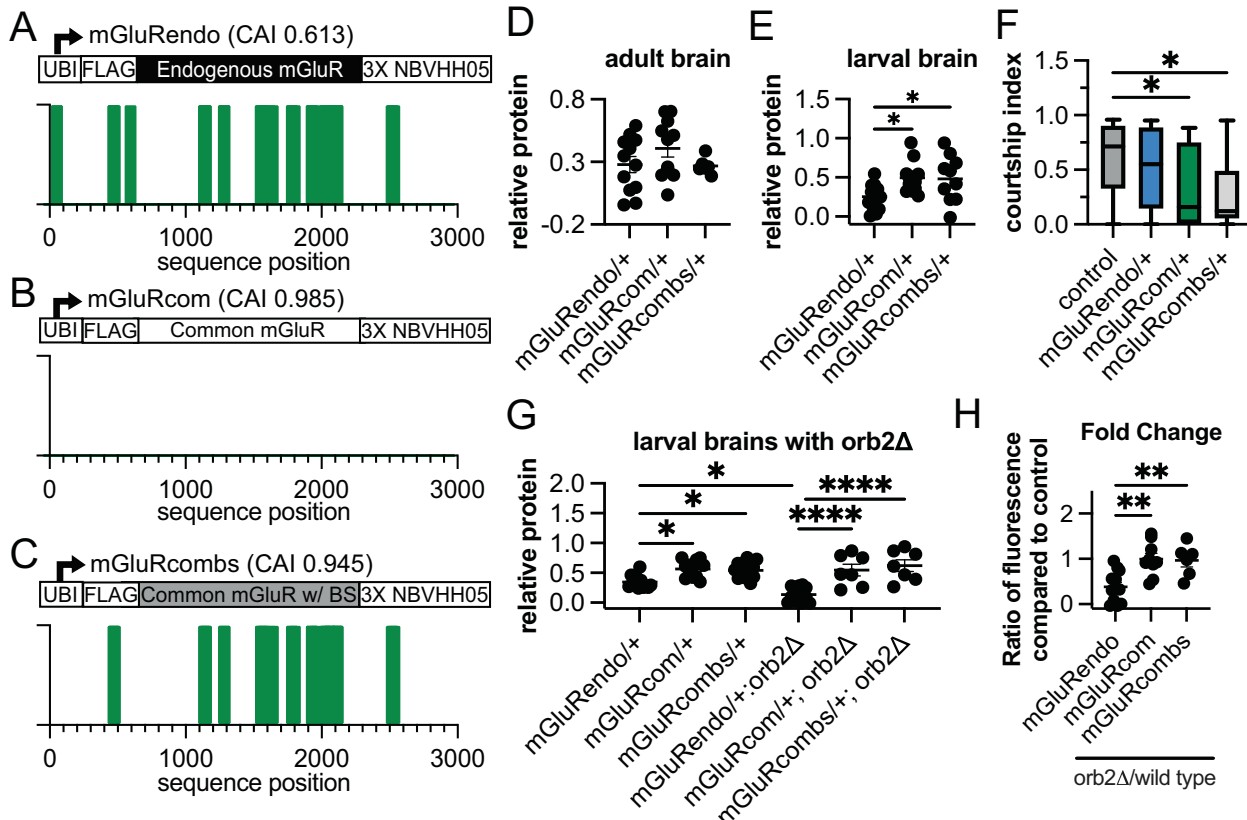

**Fig. 6 | Rare codons in mGluR, an Orb2-regulated mRNA, impact mating behavior. A, C** Schematic showing mGluR codon-modified transgenes tagged with a FLAG tag and a Vhh05 nanobody antigen for protein visualization. mGluRendo has the endogenous coding sequence, with a CAI of 0.613 (Fig. 6A), while mGluRcom has the codon optimized coding sequence, with a CAI of 0.985 (Fig. 6B), and mGluRcombs has Orb2 binding sites added back in, for a CAI of 0.945 (Fig. 6C). **D** Vhh05 protein quantification for mGluRendo heterozygote, mGluR com heterozygote, or mGluRcombs heterozygote adult male brains. Not significant by One-way ANOVA. Mean ± SEM plotted, three biological replicates conducted. Precise *N* values provided in source data file. **E** Vhh05 protein quantification of WL3 brains for mGluRendo heterozygotes, mGluRcom heterozygotes, or mGluRcombs heterozygotes. One-way ANOVA followed by Tukey's multiple comparisons test performed: mGluRendo/+ vs mGluRcom/+ *p* = 0.0241; mGluRendo/+ vs mGluRcombs/ + *p* = 0.0442. Mean ± SEM plotted, three biological replicates conducted. Precise *N* values provided in source data file. **F** Measurement of time that control or transgenic mGluR heterozygote males spent courting control females in a ten-minute

time frame (=courtship index). One-way ANOVA followed by Dunnett's multiple comparisons test performed. Precise N values provided in source data file, 5 biological replicates conducted. Control vs mGluRcom/+ *p* = 0.0273; control vs mGluRcombs/+ *p* = 0.0128. Minima, maxima, centers and bounds of box and whiskers in source data file. **G** Vhh05 protein quantification of WL3 brains for mGluRendo heterozygotes, mGluRcom heterozygotes, mGluRcombs heterozygotes, each with or without homozygous *orb2* deletion. One-way ANOVA followed by Šídák's multiple comparisons test. mGluRendo/+ vs mGluRcom/+ *p* = 0.021; mGluRendo/+ vs mGluRcombs/+ *p* = 0.0432; mGluRendo/+ vs mGluRendo/orb2Δ *p* = 0.0345; mGluRendo/orb2Δ vs mGluRcom/orb2Δ and mGluRendo/orb2Δ vs mGluRcombs/orb2Δ *p* < 0.0001. Three replicates conducted, precise N values provided in source data file. **H** Plot of fold change in Vhh05 protein expression with and without *orb2* deletion (full data presented in **G**). Ratio of Vhh05 expression in *orb2* deletion WL3 brains to average of brains with wild-type *orb2*. One-way ANOVA followed by Tukey's multiple comparisons test. mGluRendo vs mGluRcom *p* = 0.0020; mGluRendo vs mGluRcombs *p* = 0.0089.

mGluRcom or mGluRcombs between control and *orb2* deletion larval brains (Fig. 6G, H). Given that *mGluRcombs* contains 26 annotated Orb2 binding sites but is still not responsive to Orb2 regulation, we interpret our results to reflect that rare codons dampen *mGluR* mRNA expression to enable productive regulation by Orb2. Overall, our study of the neuronal Orb2-regulated gene *mGluR* reveals a role for codon-dependent regulation of protein expression in neuronal function and animal behavior.

## Discussion

In this study, we identify a fundamental change in codon usage regulation during the progression from neural stem cells to differentiated neurons. Neurons, but not NBs or ganglion mother cells, maintain an abundance of rare-codon-enriched mRNAs and produce protein from these mRNAs. This difference in codon usage regulation is significantly driven by the conserved CPEB family RNA binding protein Orb2. Orb2 is expressed at higher levels in neurons than in NBs. In neurons and the testis, the two cells/tissues most associated with Orb2

function[53–55,59,61–63], Orb2 promotes rare-codon-derived protein expression. This perfectly matches our previous discovery of the testis and brain as two tissues that distinctly upregulate rare-codon expression[16]. In the brain, we show that rare-codon-enriched reporter mRNA is stabilized by Orb2. We demonstrate that both rare-codon enrichment and a high concentration of putative Orb2 binding sites in the 3' end of the transcript are properties of numerous Orb2-upregulated brain mRNAs. Lastly, we reveal a role for rare codons in *mGluR*, an endogenous rare-codon-enriched mRNA with abundant Orb2 binding sites that is stabilized by Orb2 in the brain. Changing endogenous rare codons to common codons in an mGluR transgene increases its protein expression, prevents its regulation by Orb2, and decreases social activity in adult male flies. Given the many examples of rare codons productively fine-tuning expression of specific genes in disparate contexts[13,15,16,24,29,106,107], it is critical to identify the molecular mechanisms controlling such expression. Our findings here reveal a mechanism by which rare-codon mRNAs can be stabilized, in the context of stem cell differentiation.

## Orb2 as a molecular switch to control the stability of rare-codon-biased mRNAs

Developmental transitions such as stem cell differentiation involve alterations in cell proteomes. Recent work in this area is beginning to illuminate codon usage differences between proliferating and differentiating cell states[29,108,109] and even within stem cell lineages[18,110]. In stem cells, post-transcriptional control of gene expression is critical to promote transitions from progenitor to differentiated states[111–114]. Such regulation seems especially crucial in neural stem cells[115–117]. Here, we find that codon usage regulation is an important part of the transition from progenitor to differentiated cell expression in a defined neural stem cell lineage. Evidence suggests that other stem cell lineages are also impacted by codon-dependent regulation. For example, codon alteration of the murine KRas gene alters hematopoietic stem cell proliferation and differentiation[118]. Additionally, murine oligodendroyctes and oligodendryocyte progenitor cells have opposite codon usage preferences, which may be disrupted in leukodystrophies or white matter disease[110]. Part of this process appears to be driven by differential tRNA modification between progenitors and oligodendrocytes[110], and other studies have also found a relationship between tRNA modification and cell type-specific gene expression[40,119]. The strongest case for correlating cell states and tRNA abundances themselves appears to be with proliferating cells[112,119–121]. However, other studies have found that changing tRNA pools have no implication on development and cell identity[122–124], including in the *Drosophila* embryonic nervous system[19]. Future studies can assess how tRNA regulation functions during neuronal differentiation in the *Drosophila* brain.

Orb2 in *Drosophila* and its CPEB family orthologs in other metazoans have been heavily implicated in neurodevelopment and function[93,125]. The role that this family plays in neurons has been tied to their ability to translationally activate and repress transcripts dynamically, which is necessary for memory formation, long-term potentiation, and growth cone extension[55,59,126–129]. CPEB proteins are overwhelmingly active in the brain and germline. The brain, along with germline tissues like the testes, are well studied for unique regulation of many aspects underlying gene expression, including splicing (reviewed in[44]) and polyadenylation (reviewed in ref. [48,130]). We recently described the brain and testes as uniquely able to translate rare-codon-enriched reporters[16]. This leads us to posit that CPEB protein expression and rare-codon enrichment in target genes are necessary for function of these two types of tissues. Future study can assess if CPEB proteins play a broader role in codon-dependent expression in specific cells and tissues.

### Rare-codon enrichment allows fine-tuned regulation of neuronal transcripts like *mGluR*

Dynamic regulation of mRNA stability[126,131] and translation[132–135] are well-described phenomena in neurons. There are many cis-regulatory characteristics of an mRNA transcript that are uniquely regulated in neurons compared to other cell types. These include unique enrichment of RNA binding protein motifs in alternative last exons and 3' UTRs[136,137] and alternative transcription start and termination sites leading to neuronal-specific isoforms[138–140]. This study and others has shown that neuronally expressed mRNAs also have higher enrichment of rare codons[19,40,41]. This suggests that rare-codon enrichment could be part of a complex code controlling gene expression regulation in neurons. Another part of this code is Orb2 regulation. We find a correlation between rare-codon enrichment and Orb2 binding sites, which includes a number of neuronal-specific genes. Most of the other RNA binding proteins (ELAV/Hu RNA binding proteins[141], PUF proteins[142], Bicaudal proteins[143]) that bind U-rich elements in the 3'UTR require longer binding sites than the 6-nucleotide site bound by Orb2[85]. The other CPE binding protein in *Drosophila*, Orb, is not expressed at high levels in the brain, but is specifically expressed in the testis[85].

Therefore, Orb2 may be unique among U-rich binding RNA binding proteins in regulation of rare codons.

We posit that rare-codon enrichment is needed to keep levels of some Orb2 targets low to allow for stabilization by binding proteins such as Orb2. In line with this idea, we previously showed that the Ras GTPase is controlled by different molecular regulators when encoded by rare instead of common codons[13]. This is also supported by our results showing that common codons in *mGluR* leads to higher expression and behavioral phenotypes that cannot be rescued by Orb2 binding sites. While we do find a correlation between increased Orb2 regulation and rare-codon enrichment in our work, there is a limit as to how much either mechanism can regulate gene expression on its own. If Orb2 binding sites were sufficient to regulate gene expression of *mGluR*, adding back the Orb2 binding sites would rescue the phenotypes we see in the *mGluRcombs* background – it does not. If rare codons were sufficient to allow specific expression in the brain and testes, we would see the highest GFP protein in our reporters with the most rare codons – we do not. Neither mechanism seems dominant, and both are necessary for control of gene expression during neuronal differentiation and function.

Many of the rare-codon-enriched mRNAs that we identified here to be specifically expressed in neurons function at the synapse, including mGluR. The synapse is dynamically regulated, specifically by CPEB proteins[55,57,125,127,128], suggesting rare-codon enrichment may have evolved in these genes as a way to keep levels of expression low until they are needed. Taken together, our results present a paradigm where rare-codon enrichment within functionally related genes along with cell type and stage-specific expression of regulators cooperate to allow dynamic mRNA stability and translation control in *Drosophila* neurons.

## Methods

### Generation of codon-modified reporters in *Drosophila*

Codon-altered GFP transgenes were generated as described previously at the *attP40* (2 L) site[16]. Exon sequences for *mGluR* were downloaded from FlyBase (FB2022_03 Dmel release 6.46[144]), and codons modified according to the codon usage frequencies in the *Drosophila melanogaster* genome taken from the Kazusa codon usage database (https://www.kazusa.or.jp/codon/cgi-bin/showcodon.cgi?species=7227)[145]. Sequences were subsequently generated through gene synthesis (Twist Biosciences), cloned into a pBID-UBI plasmid (modified from Addgene plasmid #35200), transformed into NEB 5-alpha competent cells. Plasmids were purified using a ZymoPure II Plasmid Midiprep Kit (Zymo Research). Plasmids were injected into *attP40* (2 L) flies (Model System Injections). Full sequences are provided in Supplementary Dataset 1.

### Fly stocks

All flies were raised at 25 °C on standard media (Archon Scientific, Durham, NC). See Supplemental Dataset 4 for list of fly stocks used in this study and their sources. The following stocks were generated for this study: mGluRendo, mGluRcom, and mGluRcombs.

To generate MARCM clones[69], female flies containing *hs-flp*, *tub-GAL80*, and *Frt19A* on the X chromosome (Bloomington stock #5133), GFP-rare[b] on the 2nd chromosome, and UAS-myrRFP (Bloomington stock #7119) on the 3rd chromosome were crossed to males containing *Frt19A* (Bloomington stock #1744) on the X chromosome and *tub-*GAL4 (Bloomington stock #5138) on the third chromosome. Larvae were heat-shocked for 15 minutes at 37 °C either 24, 48 or 72 hours before WL3 stage and dissection for microscopy.

### Fluorescence imaging

Whole larval screening and image acquisition was on a Leica MZ10 F stereoscope (Leica Plan APO 1.0× objective #10450028) and Zeiss AxioCam MRc r2.1 camera. Larval and adult tissues were prepared for imaging with antibody staining by dissecting in 1XPBS, fixing in 1× PBS, 3.7% paraformaldehyde, and 0.3% Triton-X for 30 minutes. Primary

antibodies were dissolved in PBS with 0.03% Normal Goat Serum and 0.1% Triton-X and tissues incubated overnight at 4 °C. Primary antibodies included Dpn (1:250, Abcam), Hts (1:200, Developmental Studies Hybridoma Bank) Elav (1:500, Developmental Studies Hybridoma Bank), Wrapper (1:100, Developmental Studies Hybridoma Bank) and Repo (1:500, Developmental Studies Hybridoma Bank). Secondary antibodies (Alexa fluor 647, 1:1000, Molecular Probes) were dissolved in PBS with 0.03% Normal Goat Serum and 0.1% Triton-X with Hoechst 33, 342 (1:1500) and tissues incubated for 2 hours at room temperature. Tissues were prepared for single molecule inexpensive RNA fluorescent in situ hybridization (smiFISH) by fixation, then three subsequent washes in 0.3% Tween in PBS. They were blocked in 10% formamide in 2× Sodium Chloride- Sodium Citrate (SSC), then incubated overnight, shaking at 37 °C in 10% formamide and 10% dextran in 2× SSC with FISH probes (IDT). GFP-rare[b] and GFP-common probes were used at 1:100, *mGluR* probes were used at 1:50, *shakB* probes were used at 1:75. Probes were generated as described[80]. Sequences are listed in Supplementary Dataset 5. Nanobody detection was as described[101]. In brief, tissues were fixed and blocked as described above, then incubated in NbVHH05-HA at 1:500 overnight at 4 °C. Tissues were then washed in 1× PBS and stained with secondary antibodies (Alexa Fluor 488, 1:1000) and DAPI (1:2000) at room temperature for 2 hours. Tissues were imaged on a Nikon A1 Confocal.

### Fluorescently marked protein and RNA quantification

GFP fluorescence or nanobody fluorescence intensity was measured using the Measure tool in ImageJ. Each dot represents one animal. Unless a cell type is specified, the entire lobe of the larval brain was the region of interest. When values are presented as a ratio of control, the measurements for all control brains on a slide were averaged and each individual brain value was divided by that average value. When values are presented as relative, fluorescence intensity of values from the antennal lobe are subtracted from the fluorescence intensity of the region of interest. Line profile values were obtained using the line profile tool from ImageJ. In Fig. 2B, E, values were normalized with the average measurement for the total brain being set to 1 and all other values shown as a ratio of that.

Puncta values from single molecule inexpensive RNA fluorescent in situ hybridization (smiFISH) were counted using the Spot Counter tool on ImageJ. No prefiltering was used, box size was set to the size of 3 pixels, and noise tolerance was set to the minimum value of fluorescence for the entire image. Puncta counts were normalized to control tissues that either did not express the transgene or were only probed with secondary fluorescently labeled affinity probes. Cell types were identified using membrane-targeted fluorescent protein driven by cell type-specific GAL4 drivers. Neuroblasts and GMCs were marked by myristoylated RFP protein driven by *earmuff*-GAL4, and distinguished by size. Neurons were marked by myristoylated RFP protein driven by *elav*-GAL4.

Sidak's is most appropriate to use when comparing a set of means derived from data generated in independent measurements[146,147].

### CAI

CAI calculations for reporters and endogenous genes were performed using CAIcal[133]. A database of all annotated genes in the *Drosophila melanogaster* genome was generated using CDS sequences downloaded from Flybase (FB2017_05 Dmel release 6.18), the *Drosophila melanogaster* codon usage table from the Kazusa codon usage database[145] and the local version of CAIcal[148]. Rare-codon-enriched mRNAs were defined as CAI below 0.68, mid-range mRNAs were defined with a CAI between 0.68 and 0.78, and common-codon-enriched mRNAs were defined as having a CAI greater than 0.78.

### mRNA/protein ratio calculations

Mean relative mRNA/protein ratio was calculated by dividing the average protein abundance of GFP-rare[b] (normalized to GFP0D) by the average mRNA abundance (normalized to GFP0D) and converting to a percentage. Average values were used as protein, and mRNA abundance data were not obtained from paired samples.

### Protein preparation and analysis by western blotting

Protein samples were prepared as previously described[13]. Briefly, tissues or whole animals were homogenized in Laemmli buffer (larval tissues in 50 μl, adult tissues in 100 μl) on ice, then boiled for 5 min. Samples for measurement of GFP reporters were separated on 12% sodium dodecyl sulfate-polyacrylamide gels by electrophoresis (SDS-PAGE) at 200 V. Samples for measurement of mGluR transgenes were separated on 8% SDS-PAGE at 120 V. Proteins were transferred onto nitrocellulose membranes using an iBlot 2 Dry Blotting System (Invitrogen, Waltham, MA) set to 20 V, 6 min. The following antibodies were used: anti-FLAG M2 (1:500, Sigma, anti-mouse), anti-α-tubulin (1:2, 000, Sigma, anti-mouse), rabbit anti-GFP (1:1000, Life Technologies, #A11122). Signal was detected using a LI-COR Odyssey CLx and analyzed using Image Studio version 5.2 (LI-COR Biosciences). Band intensity for GFP and flag expression was measured and then normalized to expression of alpha tubulin as a loading control.

### Actinomycin D treatment

Twenty WL3 brains for each condition were dissected and placed into PBS. At the start of the treatment, brains were moved into either new PBS (no treatment) or PBS with 700 μM ActD (treatment) for the duration. Brains were kept on a nutator during treatment. At the end of the treatment, brains were flash-frozen. After freezing, RNA was prepared and the samples proceeded to qRT-PCR.

### RNA preparation and qRT-PCR

RNA preparation and qRT-PCR proceeded as previously described[16]. Dissections of WL3 larval brains took place under RNAse-free conditions, in PBS and in less than 2 hours. Brains were flash-frozen before RNA preparation. Brains were first homogenized frozen with no buffer, then homogenized in 50 uL Trizol reagent. Another 450 uL of Trizol reagent was added to each sample, then the manufacturer's protocol was followed. Glycogen was used as a carrier and samples were resuspended in molecular grade water at 60 degrees Celsius. NA was then treated with DNase I at room temperature for 15 min before terminating the reaction by adding 2.5 mM EDTA and incubating at 65 °C for 10 min. Quantification of RNA was performed on a Qubit 3 fluorometer and samples were diluted to match the concentration of the lowest concentration sample, then immediately proceeded to cDNA synthesis. Equal amounts of RNA for all samples directly compared to one another were simultaneously transcribed into cDNA using iScript cDNA synthesis kit (BIO-RAD, Hercules, CA, cat#170−8891) according to the manufacturer's protocol. No Reverse Transcriptase (NRT) controls were also run simultaneously for each sample to control for genomic DNA contamination. Quantitative Real-Time PCR (qRT-PCR) was run simultaneously on all samples compared to one another, corresponding NRT controls, and No Template Controls (NTC) for each primer pair using Luna Universal qCPR Master Mix (NEB, Ipswich, MA, #M3003) following the manufacturer's protocol (1 μl cDNA per 10 μl reaction). A CFX384 Touch Real-Time PCR Detection System (BIO-RAD) was used for cDNA amplification and detection of FAM/SYBR Green fluorescence. qRT-PCR run data was analyzed using BIO-RAD CFX Manager software. Primer sequences are listed in Supplementary Dataset 5. Relative mRNA levels were calculated using the $2^{\Delta CT}$ method to internally normalize the expression of reporter mRNA and *mGluR* or *ND-18* levels to a housekeeping gene, *RP49*.

## Cell-type-specific expression of RNA transcripts

Fragments per kilobase million (FPKM) counts for neurons and neuroblasts were accessed through the supplemental Dataset 1[81] and converted to transcripts per million (TPM) using the following formula: TPM = FPKM/(sum(FPKMall)*10^6). TPM counts for neurons, glia and different neuroblast types were accessed through "Supplemental Dataset 1[82]". Once TPM values were obtained, enrichment values for each gene were defined as being in the top 25% of genes expressed in the cell type of interest and in the bottom 60% of genes expressed in any other tissue analyzed. To be included in the CAI distribution of genes expressed specifically in a cell type, the gene had to meet these characteristics in both RNA sequencing datasets. This resulted in 434 genes called as enriched in neuroblasts and 424 genes called as enriched in neurons. Differences in CAI distributions are tested as significant using both a Kolmogorov–Smirnov and a two-tailed $t$ test.

## Codon-specific regulator screen

To select candidates to test as potential regulators of codon usage specific transcript expression, We chose mRNAs that were specifically enriched in expression in either neurons or neuroblasts based on previously published datasets. Candidates had to also be regulators of mRNA stability, splicing, modification, or translation.

To determine the role a gene plays in codon biased RNA expression, RNAi was used to knock down expression of candidates using *inscuteable* GAL4, which expresses throughout all neuroblast lineages in the larval brain (BDSC#8751[77]). Brains were dissected from WL3 larvae from each cross and stained with Elav antibody, then imaged on a Nikon A1 confocal. The fluorescence intensity of the reporter was calculated for one entire lobe of each brain using FIJI. Intensity values were normalized to brains with no RNAi line expressed that were mounted on the same slide. One-way ANOVA followed by Dunnett's multiple comparisons test was used to determine significant changes in fluorescent protein expression after candidate gene depletion.

## Generating RNA sequencing libraries

Protocol was followed using RNase-free reagents. Brains from WL3 of the genotypes of interest (*insc*-GAL4 x *w* RNAi and *insc*-GAL4 x *orb2* RNAi) were dissected in cold 1XPBS with RNase-out (Invitrogen) treated forceps and immediately flash-frozen, then placed at −80 °C. Dissections were done in batches of 20 in under a week to reach 60 brains per replicate, with two replicates per genotype. 500 μL of Trizol (Sigma) total was used for each replicate, with proportionate amounts added to each batch of frozen brains before homogenization. After homogenization and centrifugation at 12 kG at 10 minutes at 4 °C, the supernatant from each batch was pooled. The supernatant was incubated at 5 minutes at room temperature, then 100 μL of chloroform was added. Samples were shaken for 30 seconds and then incubated for 3 minutes at room temperature. Next, they were centrifuged at 12KG for 15 minutes at 4 °C. The supernatant was discarded, then the pellet was washed with 1 mL 70% ethanol. The samples were centrifuged at 7.5 KG for 5 minutes at 4 °C. Supernatant was discarded and the wash step was repeated. The pellet was gently dried, then 50 μL of DEPC water was added. The sample was incubated at 55 °C for 10 minutes. Then, the TruSeq stranded mRNA kit protocol (Illumina) was followed to make libraries. Libraries were sent to Novogene (Beijing, China) for sequencing on an Illumina Novaseq machine to generate paired end, 150 base pair reads.

## RNA-sequencing analysis

A total of 140 million reads (average 35 million reads per sample) were obtained. The FASTQ data were first processed with TrimGalore! to remove adapters and low quality reads. HISAT2 was used to align (about 50% of reads were aligned for each sample) and mapped to the DM.BDGP6.32.105 genome downloaded from Ensembl. The featurecounts function was used to count the reads per feature, then followed by DESeq2 to determine differential gene expression. RNA-seq data will be available in NCBI Gene Expression Omnibus.

## Annotated Orb2 binding site mapping and quantification

The last quartile of the CDS and the 3' UTR of transcripts that had significantly changed expression with depletion of *orb2* and a randomly selected group of 800 non-significantly changed transcripts were collated and queried in the RBP-map database (http://rbpmap. technion.ac.il/[92]) to identify the position and number of potential Orb2 binding motifs. We looked for binding sites in the 3' end of the CDS because there is evidence that RNA binding proteins previously indicated to only bind the 3' UTR also regulate RNA expression through the CDS[89,91,149,150]. The motif matrix used to search had been previously determined[85,151]. Stringency level settings were applied with 2 thresholds: $p$ value < 0.005 for significant hits and $p$ value < 0.01 for suboptimal hits. Number of binding sites was normalized to length of sequence queried before plotting.

## Behavioral testing

Courtship behavior was measured as described[100]. Briefly, male flies of the desired genotype were aged 4 days in isolation from female flies. They were then placed in an arena with a virgin female *w^1118* fly for 12 minutes while being video recorded. Each male fly was scored for the ratio of time of the total 10 minutes spent executing courtship behaviors (orienting, following, tapping, licking, and attempted copulation) as a proxy of social behavior. Differences in amount of time spent courting between genotypes were determined using One-Way ANOVA followed by Dunnett's multiple comparison test.

## Reporting summary

Further information on research design is available in the Nature Portfolio Reporting Summary linked to this article.

## Data availability

The data supporting the findings of this study are available from the corresponding authors upon request. The RNA sequencing data generated in this study have been deposited in the GEO database under accession code GSE263513. Previously published RNA sequencing datasets used in this study are also available on the GEO database as follows: Single-cell RNA sequencing of the brain dataset 1, Single-cell RNA sequencing of the brain dataset 2 (GSE38764), Orb2 binding targets in S2 cells (GSE59611). Source data for Figs. are provided in the source data file.

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

## Acknowledgements

The following kindly provided reagents used in this study: Bloomington Drosophila Stock Center, Developmental Studies Hybridoma Bank, Vienna Drosophila Resource Center, Dr. Kausik Si, Dr. Paul Schedl, Drosophila Genomics Resource Center. The Duke Light Microscopy Core Provided imaging support. We thank Anne West, Chris Nicchitta, Scott Allen, and Fox lab members for their comments on the manuscript. This project was supported by ACS grant RSG-128945 to D.F., NIH grants R35 GM140844 and R01 HL111527 to A.L., and NIH grants R01 GM146010 and R01 NS109401 to P.V., and NIH grant HD113543 to D.F.

## Author contributions

R.K.S. and D.T.F. conceptualized the study. R.K.S. and. P.N. acquired the findings. A.L. provided guidance on bioinformatic analyses. P.C.V. provided guidance on courtship experiments. J.K.S. provided guidance and mentorship for PN experiments. All authors contributed ideas for the design and completion of the study. R.K.S. and D.T.F. wrote the original draft, and all authors reviewed the paper.

## Competing interests

The authors declare no competing interests.
