## [Peer Review File · Nature Communications]

Orb2 enables rare-codon-enriched mRNA expression during
Drosophila neuron differentiationREVIEWER COMMENTS

Reviewer #1 (Remarks to the Author):

In this manuscript, Stewart et al examined in drosophila brain the neuro cell types that express rare-codon-enriched genes. The authors found that most differentiated larval brain cells, but not neuroblast cells, could efficiently translate rare codon-containing mRNAs. Using a small-scale RNAi knockdown screening analysis (a total of 45 genes), the authors reported that Orb2 is responsible for tissue-specific rare codon translation. Finally, the authors analyzed Orb2-regulated mRNAs and demonstrated that switching rare codons in mGluR leads to behavioral phenotypes.

The manuscript investigates an interesting phenomenon: why certain cell types could translate rare codons as efficiently as common codons. The group previously reported that both brain and testis tissues are unique in this aspect, but the underlying mechanism was elusive. In this study, the authors identified a potential regulator Orb2. Unfortunately, how Orb2 enables ribosomal translation on rare codons is still unclear. At the very least, the authors should discuss possible mechanisms. Another weakness in this study is the confusion roles of rare codons vs Orb2 binding sites on mRNAs. If the presence of rare codons is dominant, why is Orb2 important in this regulation? Alternatively, the function of Orb2 is independent of rare codons. This fundamental question was not clearly addressed in the manuscript.

Main concerns

1. Line 139 (Page 8) "There is almost 10-fold more GFP-rareb protein by fluorescence in neurons than in either NBs or GMCs (Fig2A, B)." How did the authors determine neuron, NB, and GMC regions? Did they use some marker genes like Dpn in NBs and earmuff-GAL4 in GMCs as mentioned in line 101? Just like in Figure 1D, the authors should provide the corresponding marker images used to identify the cell types.
2. Line 191 (Page 10) "Only knockdown of orb2, an RNA binding protein, significantly decreases expression of GFP-rareb without changing GFP-common expression". The authors should show that the decrease in GFP-rareb signal in orb2 knockdown is not due to the differentiation defects from NBs to neurons. Since the authors quantified the averaged GFP intensity in the entire animal, it is unclear whether the decrease in GFP signal is due to the decrease in the number of neurons or due to the lack of the ability of neurons to express GFP-rareb, as the authors expected.
3. Line 212 (Page 11) "These data suggest that the RNA binding protein Orb2 acts specifically in neurons to increase abundance of GFP-rareb mRNA and thereby increase translation into GFP-rareb protein". How do the authors distinguish mRNA stabilization from translational effect? This is because mRNA stabilization also increases the final output.
4. Have the authors tried Orb2 overexpression in neuroblasts? This experiment could help dissect rare codon vs Orb2 binding sites on mRNA translation.

Minor concern

Figure 3B legend.

I could not understand how the authors calculated the statistical significance annotated as a bracket just from the figure. Does the figure legend mean that the average of the p-values was less than 0.05? P-values are not usually averaged and should be discussed individually.

Reviewer #2 (Remarks to the Author):

In this manuscript, Stewart et al. report novel findings in the field of neural stem cell differentiation

into neurons in *Drosophila* model; how biased codon usage may play role in this critical developmental step. In particular, authors found that the stem cell to neuron differentiation is associated with protein expression from rare-codon enriched genes. The study then focuses on the role of cytoplasmic polyadenylation element binding protein (CPEB) Orb2 as a regulator of this process, which was identified via elegant two step subsequent RNAi screens. Orb2 was found to be a codon-specific regulator and that it is enriched in neurons. In addition, authors report that Orb2 can increase expression of mRNAs with both Orb2 binding sites and rare-codon bias. Furthermore, authors report that rare codons in an mRNA that is Orb2 regulated, the mGluR, can alter behavior. Overall, the conclusions stated by the authors are supported by the results presented. In sum, this study is novel and will be of high significance to number of the research areas. This reviewer has only minor concerns:

1) In Figure 2, authors should also report translation efficiency for Ganglion Mother Cell (GMCs) as they did for neuroblasts and neurons, and include adequately this result into discussions and final conclusions.

2) Authors should discuss if Orb2 binding site may be also bound by some other RNA binding proteins or if they think this site is exclusive to Orb2.

3) In the methods, authors could explain the advantage of Šidak's multiple comparison test used in their study.

Reviewer #3 (Remarks to the Author):

In this article, the authors explore the role of codon modification in the context of *Drosophila* neuron cell differentiation. They employ codon-modified reporters to demonstrate that the transition of *Drosophila* neural stem cells into neurons facilitates the expression of proteins encoded by genes enriched in rare codons. Using two reporters, one enriched in rare codons and the other not, the authors demonstrate that neurons, as opposed to neuronal progenitor cells, exhibit an upregulation of protein expression from the rare-codon-enriched reporter. This regulatory mechanism is orchestrated by the Orb2 protein, and their analysis underscores the significance of Orb2-mediated rare codon regulation in controlling the expression of at least one transgene, namely, mGluR.

Major comment:

The paper is well-written, and the authors have presented a compelling array of evidence to support their hypotheses. However, to enhance the overall persuasiveness of the results, I would suggest that they consider employing an additional method to demonstrate the differences in translation efficiency. The authors assert an increase in translation efficiency of rare codons in neurons; however, their observations are based solely on the detection of higher RNA and protein levels using IFA and FISH, that are not fully quantitative methods. To bolster their claim, it would be advisable for the authors to incorporate additional complementary techniques such as ribosome profiling in NB, GMC, and neurons. Subsequently, they could analyze the codon usage among genes exhibiting translational upregulation or repression, investigating whether it aligns with biased codon usage in neurons. Moreover, this analysis should be conducted in the absence of Orb2 to provide a more comprehensive understanding.

Minor comments:

1. Is Orb2 enriched in rare codons? Is the tRNA pool affected during neuron cell differentiation? The authors could briefly discuss this aspect in the text.

2. I kindly request that the authors specify the exact number of biological replicates in the figure legends instead of "multiple biological replicates".

3. Figure 2: Figure 2B and 2E exhibit a notable disparity in the y-axes of the bar graphs, indicating an

inconsistent representation of protein levels that could introduce bias. I kindly request the authors to make the diagrams consistent with each other.

4. Figure 2: it would be advisable to validate the finding of figure 2B-C and 2E-F using qPCR for mRNA and Western blotting for protein analysis.

5. Line 151 – 156: translation efficiency refers to the rate at which ribosomes translate a specific mRNA into a protein. Here the authors are comparing the ratio between GFP-fluorescence of two different reporters. I kindly suggest that the authors explore an alternative approach to substantiate their assertion, such as employing ribosome profiling (ribo-seq), as discussed in the major comment.

6. Could the authors please provide an annotation in Figure 2H indicating the specific CAI?

7. Figure 4: it would be beneficial for the authors to include GFP-common as a control.

8. Could the authors kindly provide a rationale for their decision not to investigate ShakB?

9. Figure 6: similarly, to figure 2, it would be advisable to validate the finding of figure 6D-E and 6 G-H using qPCR for mRNA and Western blotting for protein analysis.

RESPONSE TO REVIEWERS

Thank you to the editor and reviewers for their thorough, insightful, and positive feedback on our initial manuscript submission. For **ALL** major comments (and some minor comments where requested) we have provided **NEW DATA**. Our revised manuscript includes **21 new data panels**, and among these are **three additional experimental approaches** (western blot, quantitative RT-PCR, and mRNA stability assays).

Most helpfully, reviewer comments prompted us to conduct **new investigation** of the role of Orb2 in codon-dependent mRNA stability. Our new results described herein and in our revised manuscript **enrich our mechanistic data** regarding Orb2 function. Specifically, our revised manuscript provides **increased specificity regarding a codon-dependent Orb2 mechanism (mRNA stability)**. As such, we have revised the abstract to indicate our newfound role of Orb2 in codon-dependent mRNA stability.

Our full revisions are detailed below in the point-by-point response. These revisions, prompted by reviewers, improve the manuscript and we are very appreciative of this valuable feedback.

REVIEWER 1

Thank you to the reviewer for noting that our manuscript:

“investigates an interesting phenomenon: why certain cell types could translate rare codons as efficiently as common codons.”

Major comments

1. Line 139 (Page 8) *“There is almost 10-fold more GFP-rareb protein by fluorescence in neurons than in either NBs or GMCs (Fig2A, B).”* How did the authors determine neuron, NB, and GMC regions? Did they use some marker genes like Dpn in NBs and earmuff-GAL4 in GMCs as mentioned in line 101? Just like in Figure 1D, the authors should provide the corresponding marker images used to identify the cell types.

Response: Thank you for this helpful suggestion. As requested, we now include the marker images used during our fluorescent protein and smFISH analysis. These **NEW DATA** are in an **ENTIRELY NEW FIG S2A,B**.

2. Line 191 (Page 10) *“Only knockdown of orb2, an RNA binding protein, significantly decreases expression of GFP-rareb without changing GFP-common expression”. The authors should show that the decrease in GFP-rareb signal in orb2 knockdown is not due to the differentiation defects from NBs to neurons. Since the authors quantified the averaged GFP intensity in the entire animal, it is unclear whether the decrease in GFP signal is due to the decrease in the number of neurons or due to the lack of the ability of neurons to express GFP-rareb, as the authors expected.*

Response: We agree that this is an important point to address. We include these **NEW DATA** on counts of neuroblasts and neurons in control vs. *orb2* in the **revised Fig S3G,H**. We find no

difference in number of neuroblasts or amount of Elav positive cells in the *orb2* depleted brains, suggesting that the impact of Orb2 on rare codon reporter expression is not due to secondary defects in neuronal differentiation.

3. Line 212 (Page 11) *“These data suggest that the RNA binding protein Orb2 acts specifically in neurons to increase abundance of GFP-rare^b mRNA and thereby increase translation into GFP-rare^b protein”. How do the authors distinguish mRNA stabilization from translational effect? This is because mRNA stabilization also increases the final output.*

Response: We thank the reviewer for their interest in how Orb2 may control the stability of GFP-rare^b mRNA. To address this important point, we include **NEW DATA in revised Fig4E-G** where we quantitatively assess GFP-rare^b mRNA stability by RT-PCR in control and *orb2* brains treated with the transcriptional inhibitor actinomycin D. These new data with an **ADDED APPROACH** reveal that GFP-rare^b is sensitive to actinomycin D treatment relative to GFP-common. Further, GFP-rare^b sensitivity to actinomycin D is enhanced by loss of *orb2*, whereas GFP-common again remains stable. These results speak to the impact of Orb2 on the mRNA stability of the rare codon-enriched mRNA GFP-rare^b. These new data provide **NEW MECHANISTIC INSIGHT** with regards to a role for Orb2 in rare codon mRNA stability.

Given the importance of the reviewer comment, we also decided to go beyond the reviewer request, and also performed similar quantitative mRNA stability analysis on mGluR, to extend our findings beyond a reporter to an endogenously expressed mRNA. In these experiments we used as a negative control ND-18, an mRNA that was previously shown to be highly stable in the nervous system (Sami et al 2022) and is enriched for common codons with a CAI of 0.862. As for GFP-rare^b, we find mGluR mRNA is highly unstable relative to ND-18 mRNA, and moreover, mGluR mRNA stability is sensitive to loss of *orb2*. These **NEW DATA**, included in **revised Fig5J-L**, bolster our new conclusion that a significant mechanism by which Orb2 regulates rare codon expression is through mRNA stability control.

We also agree with the reviewer’s point that there are challenges to distinguish mRNA stabilization from a direct effect on translation. To reflect this important point, we have made revisions to the text and figures. In our revisions, we were careful when using the term “translation efficiency.” In fact, our new data led us to more specifically focus on mRNA stability rather than translation per se. We also changed the axis title of Fig2G from “translation efficiency” to the more accurate “mRNA/protein ratio.”

With our multiple text and figure revisions regarding this important point, the revised manuscript provides new mechanistic insight into the reviewer’s question about mRNA stability and more precisely addresses the impact of Orb2 on rare codon mRNA regulation.

4. Have the authors tried Orb2 overexpression in neuroblasts? This experiment could help dissect rare codon vs Orb2 binding sites on mRNA translation.

Response: We apologize to the reviewer for not making it clearer that we performed this experiment. However, this was also an experiment that we wanted to revisit using additional reagents. Whereas in the previous manuscript we individually expressed the two Orb2 isoforms (2A and 2B) in neuroblasts, in this revised version we include **NEW DATA to Fig5C** where we co-express both isoforms. This is an important addition because, for example, both isoforms are required for Orb2's control of long-term memory (Oroz et al 2020). Our new results show that even co-expression of both Orb2A and Orb2B is not sufficient to enable detectable GFP-rare^b protein expression in neuroblasts. To make this point clearer, we revised the accompanying text with the conclusion: *"in neurons **but not NBs**, Orb2 levels, Orb2 binding sites, and rare codons converge to promote rare-codon expression."*

This suggests the need for factors beyond Orb2 to enable expression of rare codons, and potentially other cell type specific mechanisms for suppressing rare codon expression. We look forward to pursuing these other components and mechanisms in future studies.

Additional reviewer comment

If the presence of rare codons is dominant, why is Orb2 important in this regulation? Alternatively, the function of Orb2 is independent of rare codons.

Response: We really appreciate this reviewer comment as it hits on an important, albeit complicated, point. Thank you for the opportunity to expand. In short, as is often the case in biology, our data show that both variables matter (i.e., neither seems dominant).

If rare codons alone were sufficient to confer Orb2 regulation, then all constructs of a certain CAI would be regulated by Orb2. While we do find a relationship between Orb2 regulation and CAI (for example GFP-common is not regulated by Orb2 but GFP-rare^b is, we also find that not all constructs with a similar CAI to GFP-rare^b have the same expression (e.g., in **Fig5A**, compare the two data points in the box- one is ubiquitous and the other, GFP-rare^b, is tissue specific).

Similarly, if binding sites alone were sufficient to confer Orb2 regulation, then all constructs of a certain number of Orb2 binding sites would show Orb2 regulation. However, as shown in **Fig 5A**, in the range of 4-8 Orb2 binding sites, numerous tested reporters show either ubiquitous, tissue-specific, or no GFP expression. Further, for our mGluR constructs, adding back binding sites does not rescue the lack of codon-dependent mGluR expression regulation by Orb2 (as shown by comparing mGluR Com to mGluR Com bs, **Fig 6G**).

To more adequately address the dual importance of binding sites and codons, we incorporate the above points in the revised text.

Minor comment

Figure 3B legend.

I could not understand how the authors calculated the statistical significance annotated as a bracket just from the figure. Does the figure legend mean that the average of the p-values was less than 0.05? P-values are not usually averaged and should be discussed individually.

Response: We agree with the reviewer- the way this was shown in the figure was unclear. We have modified **Fig 3B** to show that the last five genes on the plot (*elp1*, *CG2021*, *not1*, *ewg*, and *orb2*) all had reduced GFP^{rare}^b intensity when compared to the control (p values less than 0.05 after one-way ANOVA and Šídák's multiple comparisons test). We also now list the individual p values in the figure legend.

REVIEWER 2

Thank you to the reviewer for noting that our manuscript is
“novel and will be of high significance to number of the research areas.”

Minor comments

1. *In Figure 2, authors should also report translation efficiency for Ganglion Mother Cell (GMCs) as they did for neuroblasts and neurons, and include adequately this result into discussions and final conclusions.*

Response: We thank the reviewer for this comment. and have now made the requested change in revised **Fig 2G**. Of note, per reviewer 1 comment #3, we now display the data more accurately as “mRNA/protein” rather than “translational efficiency.”

2. *Authors should discuss if Orb2 binding site may be also bound by some other RNA binding proteins or if they think this site is exclusive to Orb2.*

Response: We thank the reviewer for this comment. Although there are other RNA binding proteins that bind to U rich sites in the 3'UTR of mRNAs, there are a few qualities that distinguish Orb2's function. Most of the other RNA binding proteins (ELAV/Hu RNA binding proteins, PUF proteins, Bicaudal proteins) that bind U-rich elements in the 3'UTR require longer binding sites than the 6-nucleotide site bound by Orb2. The other CPE binding protein in *Drosophila*, Orb, is not expressed at high levels in the brain, but is specifically expressed in the testis. Although it is possible other binding sites may overlap the Orb2 binding sites in the rare-codon-enriched target mRNAs we have identified, we do not think the specific site utilized by Orb2 is bound by other RNA binding proteins. We have added text to address this point to the discussion.

3. *In the methods, authors could explain the advantage of Šídák's multiple comparison test used in their study.*

Response: Thank you for this comment. We have added this to the methods.

REVIEWER 3

Thank you to the reviewer for noting that our manuscript is:

“well written and the authors have presented a compelling array of evidence to support their hypotheses.”

Major Comment

1A. *The authors assert an increase in translation efficiency of rare codons in neurons; however, their observations are based solely on the detection of higher RNA and protein levels using IFA and FISH, that are not fully quantitative methods.*

Response: Thank you for this valuable feedback. In response, we added three different types of quantitative **NEW DATA** to the revised manuscript:

- 1) We now include quantitative RT-PCR data in new **FigS2F** showing significantly higher expression of GFP-common compared to GFP-rare^b in larval brains.
- 2) We now include quantitative western blots for both GFP and mGluR transgene expression in **Figs S2 and S5** respectively. In **FigS2E**, we now include western blots where we compare the protein expression of GFP-rare^b and GFP-common reporters in wild type brains (with many neurons) and *brat* RNAi brains (with many neuroblasts). Using this method, we show that increasing the number of neuroblasts results in a concomitant decrease in GFP-rare^b protein expression, but no change in GFP-common protein expression. Also, in **entirely new Fig S5**, we now show quantitative western blot data that corroborates our claims with quantitative microscopy for mGluR transgenes made in Fig6.
- 3) In **new Fig S2G and H**, we now include an experiment where we use gene dosage manipulation to assess the validity of our quantitative microscopy approach. show that gene dosage correlates with protein and mRNA expression at the cellular level for GFP-rare^b using quantitative microscopy. Specifically, we repeated our quantitative microscopy methods to measure GFP-rare^b protein and mRNA in animals either heterozygous or homozygous for this transgene. These data showed that a two-fold change in gene dosage correlates with roughly two-fold changes in both protein and mRNA expression, thus validating this approach as being quite quantitative.

Given the above data, we have revised the text to reflect our quantitative measurements of mRNA and protein levels more accurately. Also, as this reviewer and also reviewer 1 (comment #3) mentioned to be careful regarding use of the term translational efficiency, we have revised the text and figures to more specifically state what was measured. These changes are outlined in reviewer 1 comment #3's response.

1B. *To bolster their claim, it would be advisable for the authors to incorporate additional complementary techniques... Moreover, this analysis should be conducted in the absence of Orb2 to provide a more comprehensive understanding.*

Response: In response, and as highlighted in the reviewer 1 comment #3 response, we indeed now include **NEW DATA in Fig 4E-G and Fig 5J-L using a complementary quantitative technique**-quantitative RT-PCR under conditions where transcription is inhibited by actinomycin D, which enables us to measure mRNA stability. These new quantitative experiments and complementary methodology, helpfully suggested by the reviewer, now provides new insight into the mRNA stability (or lack thereof) of rare codon-enriched GFP^{rare}^b, as well as the impact of *orb2* depletion under conditions where transcription of this mRNA is inhibited.

Minor comments

1. *Is Orb2 enriched in rare codons? Is the tRNA pool affected during neuron cell differentiation? The authors could briefly discuss this aspect in the text.*

Response: We thank the reviewer for their questions. Orb2 is not enriched in rare codons – it's CAI is fairly near the average of the genome at 0.744. Likely, it is expressed specifically in the brain over other tissues through a mechanism unrelated to codon usage. As far as the tRNA pool during neuron cell differentiation, this has not been precisely measured in *Drosophila*. One study has shown that there is not a large difference between the tRNAs in the central nervous system in the developing embryo and those in other tissues, despite large differences in mRNA stability (Burow et al, 2018). We have added a more thorough discussion of the state of tRNA pools during development to our discussion and plan to do our own analyses of tRNAs in the brain in the future.

2. *I kindly request that the authors specify the exact number of biological replicates in the figure legends instead of "multiple biological replicates".*

Response: This is indeed important. We have added this information.

3. *Figure 2: Figure 2B and 2E exhibit a notable disparity in the y-axes of the bar graphs, indicating an inconsistent representation of protein levels that could introduce bias. I kindly request the authors to make the diagrams consistent with each other.*

Response: As requested, we now have edited the y-axes in the bar graphs in **Figs 2B and 2E**. In order to do this, we normalized each measurement to the average measurement of total brains, which was set to 1. This allows us to preserve the visual representation of the trends of reporter protein expression across the lineage, while making the graphs consistent with each other. We also retain the original graphs in **Fig S2C and D**, in order to show the large difference in expression level of GFP-common and GFP-rare^b protein.

4. *Figure 2: it would be advisable to validate the finding of figure 2B-C and 2E-F using qPCR for*

mRNA and Western blotting for protein analysis.

Response: As mentioned in the response to reviewer #3 major comment 1A, we have conducted these analyses. In **Figure S2G-H**, in **NEW DATA** we show western blot quantification of GFP-rare^b and GFP-common in either WT brains or brains enriched in neuroblasts which corroborate our findings with quantitative microscopy. We also find that decreasing gene dosage of our rare-codon biased reporter gene by half directly corresponds with halving of the protein and mRNA abundance in each cell type as measured by our quantitative microscopy methods.

5.Line 151 – 156: translation efficiency refers to the rate at which ribosomes translate a specific mRNA into a protein. Here the authors are comparing the ratio between GFP-fluorescence of two different reporters. I kindly suggest that the authors explore an alternative approach to substantiate their assertion, such as employing ribosome profiling (ribo-seq), as discussed in the major comment.

Response: As mentioned in the response to reviewer 1 major comment 3, we have now included new data on Orb2's codon-dependent role in mRNA stability. We have conducted new analyses to measure mRNA stability of our reporter mRNA and endogenous codon biased mRNAs with and without Orb2. This shows that the rare-codon enriched reporter mRNA and the rare-codon enriched endogenous mRNA are less stable than their common-codon enriched counterparts. We look forward to further exploring the relationship between mRNA stability and translation in future analyses.

6.Could the authors please provide an annotation in Figure 2H indicating the specific CAI?

Response: Thank you for this suggestion- we have added the requested annotation.

7.Figure 4: it would be beneficial for the authors to include GFP-common as a control.

Response: Great suggestion. As requested, we now include in **FigS3F and J** measurements of GFP-common protein and mRNA using our quantitative fluorescence methods across the neuroblast lineage. There is no significant difference between cell types in wild type or *orb2* depleted brains. We also now show **NEW** qRT-PCR data to measure reporter mRNA fold change with *orb2* depletion in **FigS3I**, which corroborates our quantitative microscopy results.

8.Could the authors kindly provide a rationale for their decision not to investigate ShakB?

Response: We share your interest. Given unlimited time and funds, there are many additional genes which we could focus on, with ShakB certainly being an interesting one.

9.Figure 6: similarly, to figure 2, it would be advisable to validate the finding of figure 6D-E and 6 G-H using qPCR for mRNA and Western blotting for protein analysis.

Response: We thank the reviewer for this comment. As requested, we now provide western blot data for protein expression from the mGluR constructs **in all new Fig S5** the results of which indeed validate our findings with fluorescence-based quantitation. As we also add new quantitative RT-PCR data for mGluR mRNA and mRNA stability in **revised Fig5J-L**, we have now addressed this comment by adding new quantitative mRNA and protein measurements of mGluR to the revised manuscript.

REVIEWERS' COMMENTS

Reviewer #1 (Remarks to the Author):

In this revised manuscript, the authors have addressed concerns with additional experiments. The new data about mRNA stability is informative. The discovery of Orb2 in rare codon translation will shed lights into the tissue-specific gene expression. The revised manuscript is now suitable for publication.

Reviewer #2 (Remarks to the Author):

The authors adequately addressed concerns from this reviewer.

Reviewer #3 (Remarks to the Author):

The original manuscript underwent meticulous revision, addressing each reviewer's comments with thoroughness and precision. All feedback, including that of this reviewer and others, was thoughtfully incorporated into the revised version. Consequently, the article can therefore be published in this form.